# Speedy Performance Estimation for Neural Architecture Search

**Binxin Ru**[1] *    **Clare Lyle**[1] *    **Lisa Schut**[1]    **Miroslav Fil**[1]

**Mark van der Wilk**[2]    **Yarin Gal**[1]

[1] OATML Group, Department of Computer Science, University of Oxford, UK
[2] Department of Computing, Imperial College London, UK

## Abstract

Reliable yet efficient evaluation of generalisation performance of a proposed architecture is crucial to the success of neural architecture search (NAS). Traditional approaches face a variety of limitations: training each architecture to completion is prohibitively expensive, early stopped validation accuracy may correlate poorly with fully trained performance, and model-based estimators require large training sets. We instead propose to estimate the final test performance based on a simple measure of training speed. Our estimator is theoretically motivated by the connection between generalisation and training speed, and is also inspired by the reformulation of a PAC-Bayes bound under the Bayesian setting. Our model-free estimator is simple, efficient, and cheap to implement, and does not require hyperparameter-tuning or surrogate training before deployment. We demonstrate on various NAS search spaces that our estimator consistently outperforms other alternatives in achieving better correlation with the true test performance rankings. We further show that our estimator can be easily incorporated into both query-based and one-shot NAS methods to improve the speed or quality of the search.

## 1   Introduction

Reliably estimating the generalisation performance of a proposed architecture is crucial to the success of Neural Architecture Search (NAS) but has always been a major bottleneck in NAS algorithms [13]. The traditional approach of training each architecture for a large number of epochs and evaluating it on validation data (*full training*) provides a reliable performance measure, but requires prohibitively large computational resources on the order of thousands of GPU days [55, 39, 56, 38, 13]. This motivates the development of methods for speeding up performance estimation to make NAS practical for limited computing budgets.

A popular simple approach is *early-stopping*, which offers a low-fidelity approximation of generalisation performance by training for fewer epochs [27, 14, 25]. However, if we stop training early and evaluate the set of models on validation data, their relative performance ranking may not correlate well with the performance ranking of the fully-trained models [53], i.e. their relative performance on the test set after the entire training budget has been used. Another line of work focuses on *learning curve extrapolation* [10, 23, 3], which trains a surrogate model to predict the final generalisation performance based on the initial learning curve or meta-features of the architecture. However, the training of the surrogate often requires hundreds of fully evaluated architectures to achieve satisfactory extrapolation performance and the hyper-parameters of the surrogate also need to be optimised. Alternatively, the idea of *weight sharing* is adopted in one-shot NAS methods to speed up evaluation

---

*Equal contribution. Correspondence to `robin@robots.ox.ac.uk`.

35th Conference on Neural Information Processing Systems (NeurIPS 2021).

[36, 28, 47]. Despite leading to significant cost-saving, weight sharing heavily underestimates the true performance of good architectures and is unreliable in predicting the relative ranking among architectures [49, 52]. A more recent group of estimators claim to be zero-cost [32, 1]. Yet, their performance is often not competitive with the state of the art, inconsistent across tasks and cannot be further improved with greater training budgets.

In view of the above limitations, we propose a simple model-free method, Training Speed Estimation (TSE), which provides a reliable yet computationally cheap estimate of the generalisation performance ranking of a set of architectures. Our method is inspired by recent empirical and theoretical results linking training speed and generalisation [17, 30] and measures the training speed of an architecture by summing the training losses of the commonly-used SGD optimiser during training. We empirically show that our estimator can outperform strong existing approaches to predict the relative performance ranking among architectures, and can remain effective for a variety of search spaces and datasets. Moreover, we verify its usefulness under different NAS settings and find it can speed up query-based NAS approaches significantly as well improve the performance of one-shot and differentiable NAS. Our code is available at `https://github.com/rubinxin/TSE`.

## 2   Method

**Motivation**   The theoretical relationship between training speed and generalisation is described in a number of existing works. Stability-based generalisation bounds for SGD [17, 29] bound the generalisation gap of a model based on the number of optimisation steps used to train it. These bounds predict that models which train faster obtain a lower worst-case generalisation error. In networks of sufficient width, a neural tangent kernel-inspired complexity measure can bound both the worst-case generalisation gap and the rate of convergence of (full-batch) gradient descent [2, 6]. However, these bounds cannot distinguish between models that are trained for the same number of steps but attain near-zero loss at different rates as they ignore the training trajectory.

We instead draw inspiration from another approach which incorporates properties of the trajectory taken during SGD, seen in the information-theoretic generalisation bounds of [33] and [34]. These bounds depend on the variance of gradients computed during training, a quantity which provides a first-order approximation of training speed by quantifying how well gradient updates computed for one mini-batch generalize to other points in the training set. In view of this link, the empirical and theoretical findings in recent work [15, 43], which show that a notion of the variance of gradients over the training data is correlated with generalisation, become another piece of supporting evidence for training speed as a measure of generalisation.

Finally, [30] prove that, in the setting of linear models and infinitely wide deep models performing Bayesian updates, the marginal likelihood, which is a theoretically-justified tool for model selection in Bayesian learning, can be bounded by a notion of training speed. This notion of training speed is defined as a sum of negative log predictive likelihoods, terms that resemble the losses on new data points seen by the model during an online learning procedure. Maximising the marginal likelihood is also equivalent to minimising a PAC-Bayesian bound on the generalisation error of a model, as shown by [16]. In particular, this suggests that using the TSE approach for model selection in Bayesian models is equivalent to minimizing an estimate of a PAC-Bayes bound. Because the NAS settings we consider in our experiments use neural networks rather than Bayesian models and due to space constraints, we defer the statement and proof of this result to Appendix A.

**Training Speed Estimation**   The results described above suggest that leveraging a notion of training speed may benefit model selection procedures in NAS. Many such notions exist in the generalisation literature: [18] count the number of optimisation steps needed to attain a loss below a specified threshold, while [17] consider the total number of optimisation steps taken. Both measures are strong predictors of generalisation after training, yet neither is suitable for NAS, where we seek to stop training as early as possible if the model is not promising.

We draw inspiration from the Bayesian perspective and the PAC-Bayesian bound discussed above and present an alternate estimator of training speed that amounts to the area under the model's training curve. Models that train quickly attain a low loss after few training steps, and so will have a lower area-under-curve than those which train slowly. This addresses the shortcomings of the previous two methods as it is able to distinguish models both early and late in training.

**Definition 1** (Training Speed Estimator). *Let $\ell$ denote a loss function, $f_\theta(\mathbf{x})$ the output of a neural network $f$ with input $\mathbf{x}$ and parameters $\theta$, and let $\theta_{t,i}$ denote the parameters of the network after $t$ epochs and $i$ mini-batches of SGD. After training the network for $T$ epochs[2], we sum the training losses collected so far to get the following* Training Speed Estimate *(**TSE**):*

$$\text{TSE} = \sum_{t=1}^{T} \left[ \frac{1}{B} \sum_{i=1}^{B} \ell\left(f_{\theta_{t,i}}(\mathbf{X}_i), \mathbf{y}_i\right) \right] \tag{1}$$

*where $l$ is the training loss of a mini-batch $(\mathbf{X}_i, \mathbf{y}_i)$ at epoch $t$ and $B$ is the number of training steps within an epoch.*

This estimator weights the losses accumulated during every epoch equally. However, recent work suggests that training dynamics of neural networks in the very early epochs are often unstable and not always informative of properties of the converged networks [24]. Therefore, we hypothesise that an estimator of network training speed that assigns higher weights to later epochs may exhibit a better correlation with the true generalisation performance of the final trained network. On the other hand, it is common for neural networks to overfit on their training data and reach near-zero loss after sufficient optimisation steps, so attempting to measure training speed *solely* based on the epochs near the end of training will be difficult and likely suffer degraded performance on model selection.

To verify whether it is beneficial to ignore or downplay the information from early epochs of training, we propose two variants of our estimator. The first, **TSE-E**, treats the first few epochs as a burn-in phase for $\theta_{t,i}$ to converge to a stable distribution $P(\theta)$ and starts the sum from epoch $t = T - E + 1$ instead of $t = 1$. In the case where $E = 1$, we start the sum at $t = T$ and our estimator corresponds to the sum over training losses within the most recent epoch $t = T$.

$$\text{TSE-E} = \sum_{t=T-E+1}^{T} \left[ \frac{1}{B} \sum_{i=1}^{B} \ell\left(f_{\theta_{t,i}}(\mathbf{X}_i), \mathbf{y}_i\right) \right], \quad \text{TSE-EMA} = \sum_{t=1}^{T} \gamma^{T-t} \left[ \frac{1}{B} \sum_{i=1}^{B} \ell\left(f_{\theta_{t,i}}(\mathbf{X}_i), \mathbf{y}_i\right) \right]$$

The second, **TSE-EMA**, does not completely discard the information from the early training trajectory but takes an exponential moving average of the sum of training losses with $\gamma = 0.9$, thus assigning higher weight to the sum of losses obtained in later training epochs.

We empirically show in Section 4.2 that our proposed TSE and its variants (TSE-E and TSE-EMA), despite their simple form, can reliably estimate the generalisation performance of neural architectures with a very small training budget, can remain effective for a large range of training epochs, and are robust to the choice of hyperparameters such as the summation window $E$ and the decay rate $\gamma$. However, our estimator is *not* meant to replace the validation accuracy at the end of training or when the user can afford large training budget. In those settings, validation accuracy remains as the gold standard for evaluating the true test performance of architectures. Ours is just a speedy performance estimator for NAS, aimed at giving an indication early in training about an architecture's generalisation potential under a fixed training set-up.

Our choice of using the training loss, instead of the validation loss, to measure training speed is an important component of the proposed method. While it is possible to formulate an alternative estimator, which sums the validation losses of a model early in training, this estimator would no longer be measuring *training speed*. In particular, such an estimator would not capture the generalisation of gradient updates from one mini-batch to later mini-batches in the data to the same extent as TSE does. Indeed, we hypothesise that once the optimisation process has reached a local minimum, the sum over validation losses more closely resembles a variance-reduction technique that estimates the expected loss over parameters sampled via noisy SGD steps around this minimum. We show in Figure 1 and Appendix C that our proposed sum over training losses (TSE) outperforms the sum over validation losses (SoVL) in ranking models in agreement with their true test performance.

## 3 Related Work

Various approaches have been developed to speed up architecture performance estimation, thus improving the efficiency of NAS. Low-fidelity estimation methods accelerate NAS by using the

---

[2]$T$ can be far from the total training epochs $T_{end}$ used in complete training

validation accuracy obtained after training architectures for fewer epochs (namely early-stopping) [27, 14, 56, 53], training a down-scaled model with fewer cells during the search phase [56, 38], or training on a subset of the data [22]. However, low-fidelity estimates underestimate the true performance of the architecture and can change the relative ranking among architectures [13]. This undesirable effect on relative ranking is more prominent when the cheap approximation set-up is dissimilar to the full training procedure [53]. As shown in Fig. 1 below, the validation accuracy at early epochs of training suffers low rank correlation with the final test performance. Another class of performance estimation methods trains a regression model to extrapolate the learning curve from what is observed in the initial phase of training or predict the final test accuracy purely based on architecture structure. Regression model choices that have been explored include Gaussian processes with a tailored kernel function [41, 19], an ensemble of parametric functions [10] , tree-based models [7, 20, 50], a Bayesian neural network [23] and a $\nu$-support vector machine regressor ($\nu$-SVR)[3] which achieves state-of-the-art prediction performance [45]. Although these model-based methods can often predict the performance ranking better than their model-free early-stopping counterparts, they require a relatively large amount of fully evaluated architecture data (e.g. 100 fully evaluated architectures in [3]) to train the regression surrogate properly and optimise the model hyperparameters in order to achieve a good prediction performance. The high computational cost of collecting the training set makes such model-based methods less favourable for NAS unless the practitioner has already evaluated hundreds of architectures on the target task. Moreover, both low-fidelity estimates and learning curve extrapolation estimators are empirically developed and lack theoretical motivation.

Weight sharing is employed in one-shot or gradient-based NAS methods to reduce computational costs [36, 28, 47]. Under the weight-sharing setting, all architectures are considered as subnetworks of a supernetwork. Only the weights of the supernetwork are trained while the architectures (subnetworks) inherit the corresponding weights from the supernetwork. This removes the need for retraining each architecture during the search and thus achieves a significant speed-up. However, the weight sharing ranking among architectures often correlates poorly with the true performance ranking [49, 52, 54], meaning architectures chosen by one-shot NAS are likely to be sub-optimal when evaluated independently [54]. In Section 4.4, we demonstrate that we improve the performance of weight sharing in correctly ranking architectures by combining our estimator with it.

Recently, several works propose to estimate network performance without training by using methods from the pruning literature [1] or examining the covariance of input gradients across different input images [32]. Such methods incur near-zero computational costs but their performances are often not competitive and do not generalise well to larger search spaces, as shown in Section 4.2 below. Moreover, these methods can not be improved with additional training budget. Another line of work studies the effect of architecture encoding on speeding up the search [48, 44, 40]. However, learning better latent representation for architectures is orthogonal to the performance estimation.

Apart from the above mentioned performance estimators used in NAS, many complexity measures have been proposed to analyse the generalisation performance of deep neural networks. [18] provides a rigorous empirical analysis of over 40 such measures. This investigation finds that sharpness-based measures [31, 21, 35, 12] (including PAC-Bayesian bounds) obtain a good correlation with test set performance, but their estimation require adding randomly generated perturbations to the network parameters and the magnitude of the perturbations needs to be carefully optimised with additional training, making them unsuitable performance estimators for NAS. Optimisation-based complexity measures, which counts the number of steps required to reach a certain loss value, also perform well in predicting generalisation. However, as discussed in Section 2, it is closely related to our approach but not as easy to deploy as our estimators under the NAS setting.

## 4 Experiments

In this section, we first evaluate the quality of our proposed estimators in predicting the generalisation performance of architectures against a number of baselines (Section 4.2), and then demonstrate that simple incorporation of our estimators can significantly improve the search speed and quality of both query-based and weight-sharing NAS (Sections 4.3 and 4.4).

We measure the true generalisation performance of architectures with their final test accuracy after being completely trained for $T_{end}$ epochs. To ensure fair assessment of the architecture performance only, we adopt the common NAS protocol where all architectures searched/compared are trained

Table 1: NAS search spaces used. The true test accuracy of architectures from each search space is obtained after training with SGD on the corresponding image datasets for $T_{end}$ epochs. $N_{total}$ denotes the total possible architectures exist in the search space and $N_{samples}$ denotes the number of architectures we sample/generate for our experiments.

| Search space | $T_{end}$ | $N_{samples}$ | $N_{total}$ | Image datasets |
|---|---|---|---|---|
| NASBench-201 (NB201) [11] | 200 | 6466 | 15625 | CIFAR10, CIFAR100, ImageNet-16-120 |
| DARTS [28, 42] | 100 | 5000 | $\mathcal{O}(2^{42})$ | CIFAR10 |
| ResNet/ResNeXt [37] | 100 | 50000 | $\mathcal{O}(2^{26})$ | CIFAR10 |
| RandWiredNN (RWNN) [46, 40] | 250 | $69 \times 8$ | $\mathcal{O}(2^{378})$ | Flower102 |

and evaluated under the *same* set of hyper-parameters. Also following [50] and [11], we compare different estimators based on their Spearman's rank correlation which measures how well their predicted ranking correlates with the true test ranking among architectures.

We compare the following performance estimation methods: our proposed estimators **TSE**, **TSE-EMA** and **TSE-E** described in Section 2 and simply **the training losses at each mini batch (TLmini)**. **Sum of validation losses over all preceding epochs (SoVL)**[3] is similar to TSE but uses the *validation* losses. **Validation accuracy at an early epoch (VAccES)** corresponds to the early-stopping practice whereby the user estimates the final test performance of a network using its validation accuracy at an early epoch $T < T_{end}$. The **learning curve extrapolation (LcSVR)** method is the state-of-the-art extrapolation approach proposed in [3] which trains a $\nu$-SVR on previously evaluated architecture data to predict the final test accuracy of new architectures. The inputs for the SVR regression model comprise architecture meta-features and learning curve features up to epoch $T$. In our experiments, we optimise the SVR hyperparameters via cross-validation following [3]. Three recently proposed zero-cost baselines are also included: an estimator based on **input Jacobian covariance (JavCov)** [32] and two adapted from pruning techniques **SNIP** and **SynFlow** [1]. We also compared to XGBoost [7] and LGBoost [20] in Appendix H.

We run experiments on architectures generated from a diverse set of NAS search spaces listed in Table 1 to show that our estimators generalise well (more details are provided in Appendix B). Note $N_{samples} = 6466$ for NASBench-201 (NB201) as it is the number of unique architectures in the space. We use the architecture information released in NAS-Bench-301 [42] for DARTS and in [37] for ResNet and ResNeXt. As for RandWiredNN (RWNN) search space [46, 40], although the number of possible randomly wired architectures are immense, they are generated via a random graph generator which is defined by 3 hyperparameters. We thus uniformly sampled 69 sets of hyperparameter values for the generator and generated 8 randomly wired neural networks from each hyperparameter value, leading to $N_{samples} = 69 \times 8 = 552$. Due to space constraints, we include the results on selecting among the generator hyperparameters for RWNN in Appendix E. All experiments were conducted on an internal cluster of 16 RTX2080 GPUs.

## 4.1 Hyperparameter of TSE estimators

Our proposed TSE estimators require very few hyperparameters: the summation window size $E$ for TSE-E and the decay rate $\gamma$ for TSE-EMA, and we show empirically that our estimators are robust to these hyperparameters. For the summation window size $E$, we test different size values, $E \in [1, 10, 20, \ldots, 70]$, on various search spaces and image datasets in Appendix D and find that $E = 1$ consistently gives the best results across all cases. This, together with the almost monotonic improvement of our estimator's rank correlation score over the training budgets, supports our hypothesis discussed in Section 2 that training information in the more recent epochs is more valuable for performance estimation. Note that TSE-E with $E = 1$ corresponds to the sum of training losses over all the mini-batches in one single epoch. We also conducted an ablation study on summing training losses below one epoch, $E \in [0.1, 0.3, 0.5, 0.7]$[4], in Appendix D. We observe again that summing over the entire epoch ($E = 1$) gives the best performance; this might be because $E = 1$

---

[3]Note, we flip the sign of TSE/TSE-EMA/TSE-E/SoVL/TLmini (which we want to minimise) to compare to the Spearman's rank correlation of the other methods (which we want to maximise).

[4]For example, $E = 0.1$ corresponds to the sum of training losses over the last $10\%$ of the mini-batches in an epoch

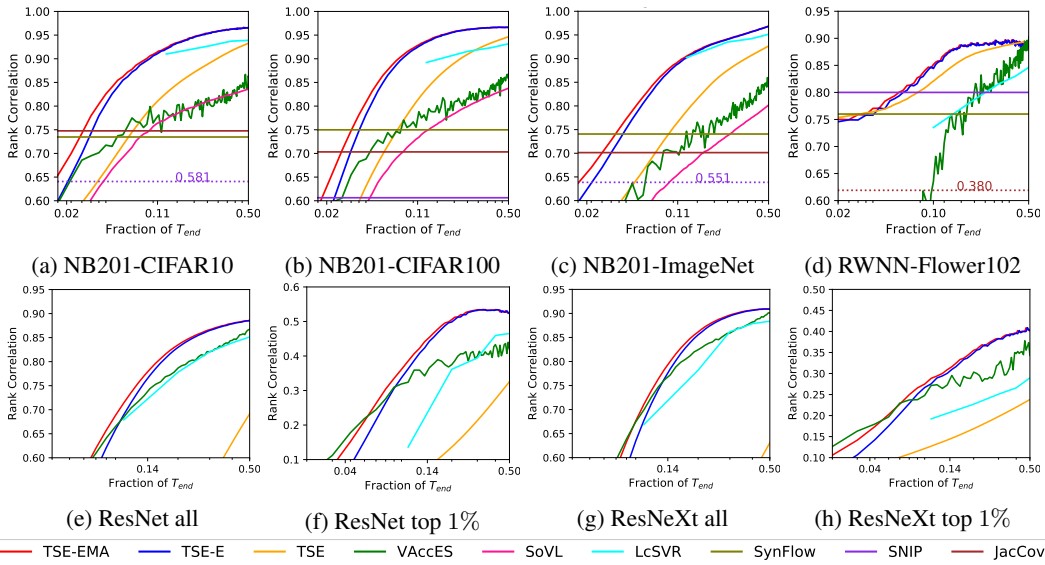

Figure 1: Rank correlation performance of various baselines for architectures from a variety of search spaces: (a) to (c) NB201 architectures on three image datasets, (d) RWNNs on Flowers102 and (e) to (h) ResNet and ResNeXt architectures on CIFAR10. In all cases, our TSE-EMA and TSE-E achieve superior rank correlation with the true test performance in much fewer epochs than other baselines. In (f) and (g), we evaluate estimators on the top 1% of the ResNet/ResNeXt architectures and show that our TSE-EMA and TSE-E can remain competitive on ranking among top architectures, which are particularly desirable for NAS. In (a) and (c), we mark SNIP in a violet dotted line labelled with its rank correlation value as it falls out of the plotted range.

covers the entire training set as is done by the Bayesian marginal likelihood and PAC-Bayes bounds, which are the theoretical inspirations for our method. On the other hand, we observe that $E \geq 0.3$ can achieve relatively close performance as $E = 1$, thus motivating our use of a small number of mini-batches to estimate TSE in one-shot and gradient-based NAS in Section 4.4.

As for $\gamma$, we show in Appendix D that TSE-EMA is robust to a range of popular choices $\gamma \in [0.9, 0.95, 0.99, 0.999]$ across various datasets and search spaces. Specifically, the performance difference among these $\gamma$ values are almost indistinguishable compared to the difference between TSE-EMA and TSE-E. Thus, we set $E = 1$ and $\gamma = 0.999$ in all the following experiments and recommend them as the default choice for potential users who want to apply TSE-E and TSE-EMA on a new task without additional tuning.

## 4.2 Comparison of Performance Estimation Quality

**Robustness across different NAS search spaces** We now compare our TSE estimators against a variety of other baselines. To mimic the realistic NAS setting [13], we assume that all of the estimators can only use the information from early training epochs and limit the maximum budget to $T \leq 0.5T_{end}$ in this set of experiments. This is because NAS methods often need to evaluate hundreds of architectures or more during the search and thus rarely use evaluation budget beyond $0.5T_{end}$ so as to keep the search cost practical/affordable. The results on a variety of the search spaces are shown in Fig. 1. Our proposed estimator TSE-EMA and TSE-E, despite their simple form and cheap computation, outperform all other methods under limited evaluation budget $T < 0.5T_{end}$ for all search spaces and image datasets. They also remain very competitive on ranking among the top 1% ResNet/ResNeXt architectures as shown in Fig. 1(f) and (g). This is particularly desirable for NAS for which we need to distinguish not only the good architectures from the bad ones but more importantly the top architectures from the merely good ones.

It is worth highlighting that TSE-EMA achieves superior performance over TSE-E especially when $T$ is small. This suggests that although the training dynamics at early epochs might be noisy, they still carry some useful information for explaining the generalisation performance of the network.

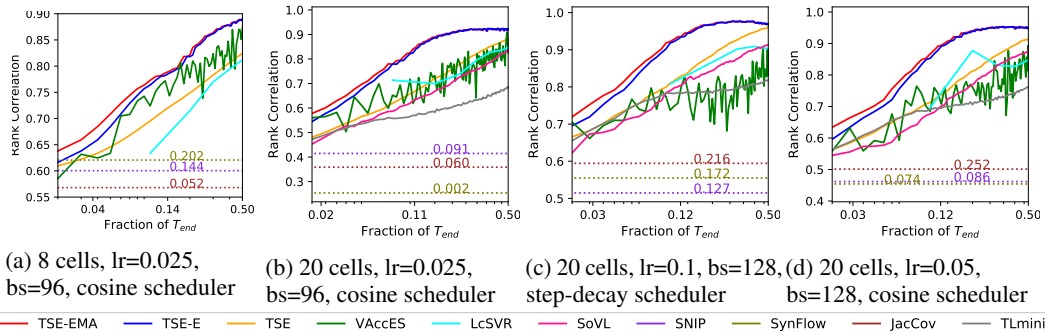

(a) 8 cells, lr=0.025, bs=96, cosine scheduler

(b) 20 cells, lr=0.025, bs=96, cosine scheduler

(c) 20 cells, lr=0.1, bs=128, step-decay scheduler

(d) 20 cells, lr=0.05, bs=128, cosine scheduler

Figure 2: Rank correlation performance of various baselines for 5000 small 8-cell architectures (a) and 150 large 20-cell architectures (b) to (d) from DARTS search space on CIFAR10. We use NAS-Bench-301 dataset(NAS301) for computing (a) and for large architectures, we test three training hyperparameter set-ups with different initial learning rates, learning rate schedulers and batch sizes as denoted in the subcaptions. On all four settings, our TSE-E again consistently achieves superior rank correlation in fewer epochs than other baselines. Note all three zero-cost estimators perform poorly (below the plotted range) on DARTS search space across all settings. We denote them in dotted lines with their rank correlation value labelled.

The learning curve extrapolation method, LcSVR, is competitive. However, the method requires 100 fully trained architecture data to fit the regression surrogate and optimise its hyperparamters via cross validation; a large amount of computational resources are needed to collect these training data in practice. The zero-cost measures JacCov and SynFlow achieve good rank correlation at initialisation but are quickly overtaken by TSE-EMA and TSE-E once the training budget exceeds 6-7 epochs. SNIP performs poorly and falls out of the plot range in Fig. 1 (a) and 1 (c).

We further validate on the architectures from the more popular search space used in DARTS. One potential concern is that if models are trained using different hyperparameters that influence the learning curve (e.g. learning rate), the prediction performance of our proposed estimators will be affected. However, this is not a problem in NAS because almost all existing NAS methods [11, 50, 46, 28, 42, 45] search for the optimal architecture under a fixed set of training hyperparameters. We also follow this *fixed-hyperparamter* set-up in our work. Verifying the quality of various estimators for predicting the generalisation performance across *different hyperparameters* lies outside the scope of this paper but present an interesting direction for future work.

**Robustness across different NAS set-ups** Here, we conduct experiments to verify the robustness of our estimators across different NAS set-ups. The relative test performance ranking among the same set of architectures can vary across different set-ups [49]. On top of the architecture data from NAS-Bench-301 [42], we also generate several additional architecture datasets; each dataset correspond to a different set-up (e.g. different architecture depth, initial learning rate, learning rate scheduler and batch size) and contains 150 large 20-cell architectures which are randomly sampled from the DARTS space and evaluated on CIFAR10. The results in Fig. 2 show that our estimator consistently outperforms all the competing methods in comparing architectures under different NAS set-ups. Note here the curve of **TLmini** corresponds to the *average* rank correlation between final test accuracy and the mini-batch training loss over the epoch. The clear performance gain of our TSE estimators over TLmini supports our claim that it is the sum of training losses, which measures the training speed and thus carries the theoretical motivations explained in Section 2, instead of simply the training loss at a single mini-batch, that gives a good estimation of generalisation performance. Note the rank correlation of all zero-cost measures drop significantly (e.g. SynFlow drops from 0.74 on NB201 to below 0.2) on the DARTS search space, and even do worse than the training losses at the first few mini-batches (TLmini at T=1). Also zero-cost measures do not take into account the training set-up when estimating architecture performance and thus their rank correlation with the true test accuracy varies largely across the set-ups. Such inconsistent prediction performance, especially given the measures' weak performance on the more practical search space, might be undesirable for real-world NAS applications.

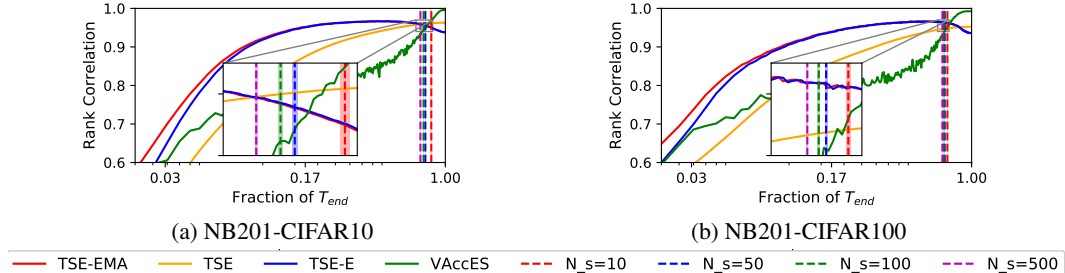

(a) NB201-CIFAR10           (b) NB201-CIFAR100

Figure 3: Rank correlation performance up to $T = T_{end}$. If the users want to apply our estimators for large training budget, they can estimate the effective range of our estimators based on the minimum epoch $T_o$ when overfitting happens among the $N_s$ observed architectures. They can then stop our estimators early at $0.9T_o$(marked by vertical lines) or switch back to validation accuracy beyond that.

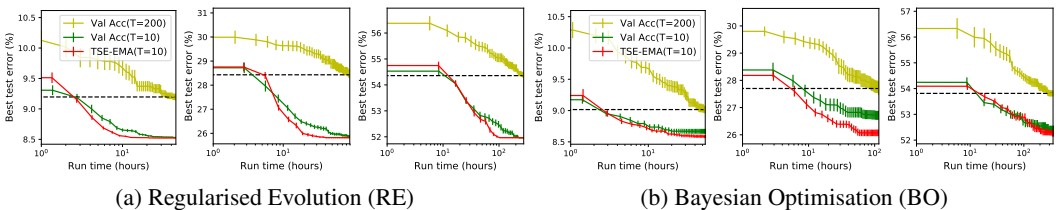

(a) Regularised Evolution (RE)           (b) Bayesian Optimisation (BO)

Figure 4: NAS performance of RE and BO in combined with final validation accuracy Val Acc (T=200), early-stopping validation accuracy Val Acc (T=10) and our estimator TSE-EMA(T=10) on NB201. For each subplot, we experiment on the three image datasets: on CIFAR10 (left), CIFAR100 (middle) and ImageNet (right). TSE-EMA leads to the fastest convergence to the top performing architectures in all cases. The black dashed line is to facilitate the comparison of runtime taken to reach a certain test error among different variants.

**Procedure to decide the effective training budget**    We include two examples showing the performance of our estimator for training budgets beyond $0.5T_{end}$ in Fig. 3. Our estimators can outperform the early-stopping validation accuracy for a relatively wide range of training budgets. Although they will eventually be overtaken by validation accuracy as the training budget approaches $T_{end}$ as discussed in Section 2, the large training budget regime is less interesting for NAS where we want to maximise the cost-saving by using performance estimators. However, if the user wants to apply our estimators with a relatively large training budget, we propose a simple method here to estimate when our estimators would be less effective than validation accuracy. We notice that that our estimators, TSE-EMA and TSE-E, become less effective when the architectures compared start to overfit because both of them rely heavily on the lastest-epoch training losses to measure training speed, which is difficult to estimate when the training losses become too small. Thus, if we observe one of the architectures compared overfits beyond $T_o < T_{end}$, we can stop the computation of TSE-E and TSE-EMA early by reverting to a checkpoint at $T = 0.9T_o$. We randomly sample $N_s = 10, 50, 100, 500$ architectures out of all architectures compared and assume that we have access to their full learning curves. We then decide the threshold training budget $0.9T_o$ (vertical lines) as the minimum training epoch that overfitting happens among these $N_s$ architectures. We repeat this for 100 random seeds and plot the mean and standard error of the threshold for each $N_s$ in Fig. 3. It is evident that we can find a quite reliable threshold with a sample size as small as $N_s = 10$. Please refer to Appendix F for more analyses.

### 4.3 Speed up Query-based NAS

In this section, we demonstrate the usefulness of our estimator for NAS by incorporating TSE-EMA, at $T = 10$ into several query-based NAS search strategies: Regularised Evolution [38] (a) in Fig. 4), Bayesian Optimisation [5] (b) in Fig. 4) and Random Search [4] (Appendix G). We perform architecture search on NB201 search space. We compare this against the other two benchmarks which use the final validation accuracy at $T = T_{end} = 200$, denoted as Val Acc (T=200) and the early-stop validation accuracy at $T = 10$, denoted as Val Acc (T=10), respectively to evaluate the

Table 2: Results of performance estimators in one-shot NAS setting over 3 supernetwork training initialisations. For each supernetwork, we randomly sample 500 random subnetworks for DARTS and 200 for NB201, and compute their TSE, Val Acc after inheriting the supernetwork weights and training for $B$ additional mini-batches. Rank correlation measures the estimators' correlation with the rankings of the true test accuracies of subnetworks when *trained from scratch independently*, and we compute the average test accuracy of the top 10 architectures identified by different estimators from all the randomly sampled subnetworks.

| B | Estimator | Rank Correlation | | | | Average Accuracy of Top 10 Architectures | | | |
| | | NB201-CIFAR10 | | | DARTS | NB201-CIFAR10 | | | DARTS |
| | | RandNAS | FairNAS | MultiPaths | RandNAS | RandNAS | FairNAS | MultiPaths | RandNAS |
|---|---|---|---|---|---|---|---|---|---|
| 100 | TSE | **0.70 (0.02)** | **0.84 (0.01)** | **0.83 (0.01)** | **0.30(0.04)** | **92.67 (0.12)** | **92.7 (0.1)** | **92.63 (0.12)** | **93.64(0.04)** |
| | Val Acc | 0.44 (0.15) | 0.56 (0.17) | 0.67 (0.05) | 0.11(0.04) | 91.47 (0.31) | 91.73 (0.21) | 91.77 (0.78) | 93.20(0.04) |
| 200 | TSE | **0.70 (0.03)** | **0.850 (0.01)** | **0.83 (0.01)** | **0.32(0.04)** | **92.70 (0.00)** | **92.77 (0.06)** | **92.73 (0.06)** | **93.55(0.04)** |
| | Val Acc | 0.41 (0.10) | 0.56 (0.17) | 0.53 (0.11) | 0.09(0.02) | 91.53 (0.55) | 92.40 (0.10) | 92.23 (0.23) | 93.34(0.02) |
| 300 | TSE | **0.71 (0.03)** | **0.851 (0.00)** | **0.82 (0.01)** | **0.34(0.04)** | **92.70 (0.00)** | **92.77 (0.06)** | **92.70 (0.00)** | **93.65(0.04)** |
| | Val Acc | 0.44 (0.04) | 0.62 (0.08) | 0.59 (0.71) | 0.06(0.02) | 91.20 (0.35) | 92.10 (0.50) | 91.43 (0.72) | 93.31(0.02) |

architecture's generalisation performance. All the NAS search strategies start their search from 10 random initial data and are repeated for 20 seeds. The mean and standard error results over the search time are shown in Fig. 4. By using our estimator, the NAS search strategies can find architectures with lower test error given the same time budget or identify the top performing architectures using much less runtime as compared to using final or early-stopping validation accuracy. The gain of using our estimator is more significant for NAS methods performing both *exploitation* and exploration (RE and BO) than that doing pure exploration (Random Search in Appendix G).

## 4.4 Improving One-shot and Gradient-based NAS

Different from query-based NAS strategies, which evaluate the architectures queried by training them independently from scratch, another popular class of NAS methods use weight sharing to accelerate the evaluation of the validation performance of architectures (subnetworks) and use this validation information to select architectures or update architecture parameters. Here we demonstrate that our TSE estimator can also be a plug-in replacement for validation accuracy or loss used in this family of NAS methods to improve their search performance.

**One-shot NAS** We first experiment on a classic one-shot method, RandNAS [26], which trains a supernetwork by uniform sampling, then performs architecture search by randomly sampling subnetworks from the trained supernetwork and comparing them based on their validation accuracy. We follow the RandNAS procedure for the supernetwork training but modify the search phase: for each randomly sampled subnetwork, we train it for $B$ additional mini-batches after inheriting weights from the trained supernetwork to compute our TSE estimator. Note that although this introduces some additional cost, our estimator eliminates the cost of evaluation on the validation set as it doesn't require validation data. Specifically, for a single DARTS architecture evaluated on CIFAR10, the RandNAS protocol takes on average 6.6 seconds to compute the validation accuracy on entire validation set, and 7.5 seconds to train for $B = 100$ additional mini-batches to compute our TSE estimator. On NB201, the recommended protocol for CIFAR10 takes on average 6.4 seconds per architecture to compute validation accuracy and only 4.4 seconds to train for $B = 100$ additional mini-batches.

In our experiments, to ensure fair comparison, we recompute the validation accuracy of each subnetwork after the additional training. We also experiment with more advanced supernetwork training techniques such as FairNAS [9] and MultiPaths [51] and show that our estimators can be applied on top of these techniques to further improve the rank correlation performance.

We evaluate the rank correlation performance and average test accuracy of the top-10 architectures recommended by different performance estimators among 500 random subnetworks sampled from the DARTS supernet[5] and 200 random subnetworks from the NB201 supernet. We repeat the experiments

---

[5]We use NAS-Bench-301 to compute the true test accuracy of each subnetwork when trained independently from scratch.

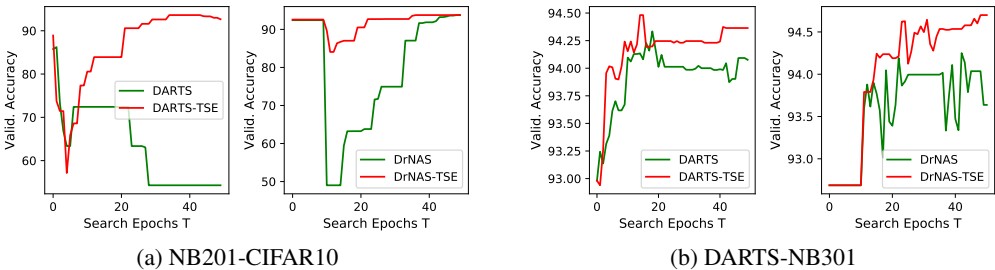

|     |     |
| --- | --- |
| (a) NB201-CIFAR10 | (b) DARTS-NB301 |

Figure 5: Test accuracy of the subnetwork recommended by differentiable NAS methods over the search epochs. Original DARTS, DrNAS (in green) use the gradient of validation loss to update the architecture parameters but their variants (DARTS-TSE and DrNAS-TSE) (in red) uses that of our estimator computed over 100 mini-batches. TSE help mitigate the overfitting of DARTS on NB201.

on $B = 100, 200, 300$ and over 3 different supernetworks training seeds. The mean and standard deviation results are shown in Table 2. It is evident that our TSE leads to $170\%$ to $300\%$ increase in rank correlation performance compared to validation accuracy and achieves higher average test accuracy of the top 10 architectures across all supernetwork training techniques and search spaces. This implies that our estimator can lead to architectures with better generalisation performance under the popular weight-sharing setting. We include the results for more estimators in Appendix G.

**Differentiable NAS** Finally, we briefly demonstrate the use of our estimators on differentiable NAS. We modify two differentiable approaches, DARTS [28] and DrNAS [8] by directly replacing the derivative of the validation loss with that of our TSE estimator computed over 100 mini-batches (B=100 as in one-shot NAS setting above) to update the architecture parameters. We include in Appendix G additional details of our adaptation. We test both approaches and their TSE variants on the NB201-CIFAR10 as well as DARTS search space. Again we use NAS-Bench-301 to obtain the true test performance of searched DARTS architectures on CIFAR10 (DARTS-NB301). The test accuracy of the subnetwork recommended over each search epoch is shown in Figure 5: we average over 3 seeds for NB201-CIFAR10 and use the default seed for DARTS-NB301. The results show that a simple integration of our estimator into the differentiable NAS framework can lead to clear search performance improvement and even mitigate the overfitting to skip-connections problem suffered by DARTS on the NB201 search space (a). The results for DARTS architectures found with our TSE estimator but retrained under the full DARTS training protocol [28] are shown in Appendix G, which again show the benefit of using our estimator.

## 5 Conclusion

We propose a simple yet reliable method for estimating the generalisation performance of neural architectures based on their training speed as measured by the sum of early training losses. Our estimator is theoretically motivated by the connection between training speed and generalisation, and outperforms other efficient estimators in terms of rank correlation with the true test performance under different search spaces as well as different training set-ups. Moreover, it can lead to significant speed-ups and performance gains when applied to different NAS strategies including one-shot and differentiable methods. We believe our estimator can be a very useful tool for achieving efficient neural architecture search. Our estimators, by reducing the computation and time required for performance evaluation during NAS, can significantly reduce the environmental costs incurred by NAS as AutoML becomes more widely used in industry. These efficiency gains can further enable potential users with limited computation budgets to use NAS methods.

## Acknowledgments and Disclosure of Funding

Binxin Ru was supported by Clarendon Scholarship. Clare Lyle was supported by the Open Philanthropy Foundation AI Fellows program. Lisa Schut was supported by EPSRC (grant number EP/S024050/1) and DeepMind. Computational resources were supported by the Oxford-Man Institute of Quantitative Finance as well as JADE HPC at the University of Oxford.

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
