# Appendix of Speedy Performance Estimation for Neural Architecture Search

Binxin Ru[1] *    Clare Lyle[1] *    Lisa Schut[1]    Miroslav Fil[1]

Mark van der Wilk[2]    Yarin Gal[1]

[1] OATML Group, Department of Computer Science, University of Oxford, UK
[2] Department of Computing, Imperial College London, UK

## A    PAC-Bayesian Generalisation Bounds Estimator

**Corollary 1** (Corollary 2 of [8]). *Given a data distribution $\mathcal{D}$, a model parameter set $\Theta$, a prior distribution $P(\theta)$ over $\Theta$, $\delta \in (0, 1]$, if the negative log likelihood $\ell$ lies within the range $[a, b]$, we have, with probability at least $1 - \delta$ over the choice of $(\mathbf{X}, \mathbf{y}) \sim \mathcal{D}^n$, which denotes $n$ input-output pairs sampled from a data distribution,*

$$\mathbb{E}_{\theta \sim \rho} \mathbb{E}_{x,y \sim \mathcal{D}}[\ell(f_\theta(x), y)] \leq a + c\left[1 - e^a (Z_{\mathbf{X},\mathbf{y}} \delta)^{\frac{1}{n}}\right], \tag{1}$$

*where $\rho$ is the posterior of model parameters $P(\theta|\mathbf{X}, \mathbf{y})$, $Z_{\mathbf{X},\mathbf{y}}$ is the marginal likelihood and $c = \frac{b-a}{1 - e^{a-b}}$.*

**Theorem 1** (Theorem 3.3 of [13]). *Let $(\mathbf{X}, \mathbf{y}) \sim \mathcal{D}^n$ denote $n$ input-output pairs sampled from a data distribution, $\theta_k$ be generated by Bayesian updates, we can obtain the following lower bound on the log marginal likelihood $\log Z_{\mathbf{X},\mathbf{y}}$:*

$$\log Z_{\mathbf{X},\mathbf{y}} \geq \sum_{k=1}^{n} \mathbb{E}_{\theta_k \sim P(\cdot|\mathbf{X}_{<k}, \mathbf{y}_{<k})} \left[\log P(x_k, y_k|\theta_k)\right] \tag{2}$$

$$\gtrapprox \sum_{k=1}^{n} \log P(x_k, y_k|\hat{\theta}_k) \tag{3}$$

$$= -\sum_{k=1}^{n} \ell\left(f_{\hat{\theta}_k}(x_k), y_k\right). \tag{4}$$

Equation 2 to Equation 3 is obtained by using MC approximation to estimate the expectation and Equation 3 to Equation 4 is because $\ell$ denotes negative log likelihood.

By substituting Equation 4 into Equation 1, we obtain a PAC-Bayesian bound, which depends on a sum of negative log likelihoods, described in Section 2:

$$\mathbb{E}_{\theta \sim \rho} \mathbb{E}_{x,y \sim \mathcal{D}}[\ell(f_\theta(x), y)] \leq a + c\left[1 - e^a (Z_{\mathbf{X},\mathbf{y}} \delta)^{\frac{1}{n}}\right]$$

$$\approx a + c\left[1 - e^a \left(e^{-\sum_{k=1}^{n} \ell(f_\theta(x_k), y_k)} \delta\right)^{\frac{1}{n}}\right]. \tag{5}$$

---

*Equal contribution. Correspondence to `robin@robots.ox.ac.uk`.

35th Conference on Neural Information Processing Systems (NeurIPS 2021), Sydney, Australia.

# B    Hardware and Datasets Description

All experiments are performed with an internal cluster of 16 RTX2080 GPUs and a Intel Core i5 CPU. The datasets we experiment with are:

- **NASBench-201 (NB201)** [7]: the dataset contains information of 15,625 different neural architectures, each of which is trained with SGD optimiser and evaluated on 3 different datasets: CIFAR10, CIFAR100, IMAGENET-16-120 for 3 random initialisation seeds. The training accuracy/loss, validation accuracy/loss after every training epoch as well as architecture meta-information such as number of parameters, and FLOPs are all accessible from the dataset. The search space of the NASBench-201 dataset is a 4-node cell and applicable to almost all up-to-date NAS algorithms. The dataset is available at `https://github.com/D-X-Y/NAS-Bench-201` with MIT License.

- **RandWiredNN**: we produce this dataset by generating 552 randomly wired neural architectures from the random graph generators proposed in [24] and evaluate their performance on the image dataset FLOWERS102 [15]. We explore 69 sets of hyperparameter values for the random graph generators and for each set of hyperparameter values, we sample 8 randomly wired neural networks from the generator. A randomly wired neural network comprises 3 cells connected in sequence and each cell is a 32-node random graph. The wiring/connection within the graph is generated with one of the three classic random graph models in graph theory: Erdos-Renyi (ER), Barabasi-Albert (BA) and Watt-Strogatz (WS) models. Each random graph model has 1 or 2 hyperparameters that decide the generative distribution over edge/node connection in the graph. All the architectures are trained with SGD optimiser for 250 epochs and other training set-ups follow those in [12]. This dataset allows us to evaluate the performance of our simple estimator on hyperparameter/model selection for the random graph generator. We will release this dataset after paper publication.

- **DARTS**: DARTS search space [12] is more general than those of NASBench-201 and contains over $10^{18}$ architectures. It's also the most widely adopted space in NAS [29, 12, 5, 25, 26, 18, 11, 16, 19, 28]. Particularly, this search space comprises a cell of 7 nodes: the first two nodes in cell $k$ are the input nodes which equals to the outputs of cell $k-2$ and cell $k-1$ respectively. The last node in the cell $k$ is the output node which gives a depthwise concatenation of all the intermediate nodes. The remaining four intermediate nodes are operation nodes take can take one out of eight operation choices. An architecture from this search space is formed by stacking the cell 8 or 20 times. We generate three DARTS architecture datasets; each dataset contains 150 20-cell architectures randomly sampled from the search space but follows a different training set-up. Specifically, for the dataset used in **Figure 2 (b) of main text**, we use an initial learning rate of 0.025, a cosine-annealing schedule and a batch size of 96 (i.e. the setting for CIFAR10 complete training in [12]. For the dataset used in **Figure 2 (c) of main text**, we use an initial learning rate of 0.1, a step-decay schedule and a batch size of 128 (i.e. the setting for ImageNet complete training in [12]). Finally, for the dataset used in **Figure 2 (d) of main text**, we use an initial learning rate of 0.05, a cosine-annealing schedule and a batch size of 128, modified from the setting in Figure 2 (b). The other training setups are the same: all architectures are trained for 150 epochs using SGD optimiser with momentum of 0.9 and regularised using a Cut-Out of 16 and a DropPath with probability of 0.2 following [12]. For these datasets, we also record the training loss for each minibatch (TLmini) on top of the conventional training and validation loss/accuracies. The minibatch training loss is used to verify our claim that it is the sum of training losses, which has nice theoretical motivation, instead of training loss itself that gives good correlation with the generalisation performance of the architectures.

- **NASBench-301 (NB301)** [20]: To experiment on a larger number of architectures formed of DARTS cells, we further experiment on this dataset which contains 23000 8-cell architectures drawn from the DARTS search space and evaluated on CIFAR10. These architectures are much smaller than the 20-cell counterparts we generate and can be trained to convergence in fewer epochs. Each architecture in this dataset is trained for 100 epochs using a SGD optimiser with an initial learning rate of 0.025, a cosine annealing schedule and a batch size of 96. The training also adopts regularisation techniques such as an auxiliary tower with a weight of 0.4 and DropPath with probability of 0.2. For these architectures, we can assess their training accuracy/loss, validation accuracy after every training epoch from the

dataset. Moreover, this dataset also provides a well trained surrogate model which can accurately predict the final test accuracy of any other possible architectures formed by 8 cells from the DARTS search space. We use this surrogate model to predict the ground-truth test accuracy of the subnetworks, when being trained from scratch independently, in the one-shot experiments in Section 4.4. The dataset is available at `https://github.com/automl/nasbench301` with Apache-2.0 License.

- **ResNet** [17]: It features two ResNet model families: ResNet and ResNeXt. The number of architecture samples as well as the architecture parameters and their corresponding range are shown in Table 1. Each architecture is trained on CIFAR10 for 100 epochs using SGD with an initial learning rate of 0.1, a cosine annealing schedule and a batch size of 128. The dataset is available at `https://github.com/facebookresearch/nds` with MIT License.

Table 1: Search spaces for ResNet and ResNeXt. Each network is formed of three stages and for each of the stage $i$, there are $d_i$ the number of blocks, $w_i$ number of channels per block. For ResNeXt, we also need to decide on the bottleneck width ratio $r_i$ and the number of groups $g_i$ per stage. The total number of possible architectures is $(dw)^3$ and $(dwrg)^3$ for ResNet and ResNeXt.

|            | $d_i$   | $w_i$      | $r_i$  | $g_i$   | $N_{samples}$ | $N_{total}$ |
|------------|---------|------------|--------|---------|---------------|-------------|
| ResNet     | [1,24]  | [16,256]   |        |         | 25000         | 1259712     |
| ResNeXt-A  | [1,16]  | [16,256]   | [1,4]  | [1,4]   | 25000         | 11,390,625  |
| ResNeXt-B  | [1,16]  | [64,1024]  | [1,4]  | ]1,16]  | 25000         | 52,734,375  |

## C  Training Losses vs Validation Losses

### C.1  Compute TSE with training losses or validation losses

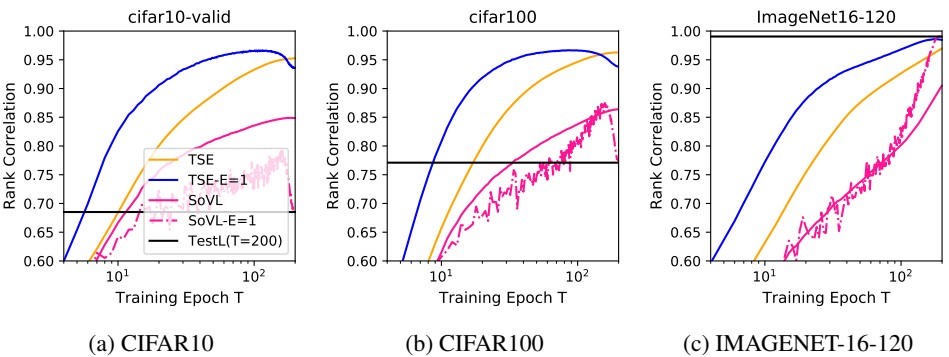

(a) CIFAR10      (b) CIFAR100      (c) IMAGENET-16-120

Figure 1: Rank correlation (with final test *accuracy*) performance of TSE (yellow) and TSE-E (blue), and those of validation losses (pink), SoVL (solid) and SoVL-E (dash dot), as well as that of final test loss (black) for architectures in NB201 on three image datasets. Note the correlation performances of final test loss and SoVL-E near the end of training get surprisingly poor for CIFAR10/100. We explain this in Appendix C.1.

We perform a simple sanity check against the validation loss on NB201 datasets. Specifically, we compare our proposed estimators, TSE and TSE-E, computed with training losses against two equivalent variants of validation loss-based estimators: Sum of validation losses (SoVL) and that over the most recent epoch (SoVL-E with $E = 1$). For each image dataset, we randomly sample 5000 different neural network architectures from the search space and compute the rank correlation between the true test accuracies (at $T = 200$) of these architectures and their corresponding TSE/TSE-E as well as SoVL/SoVL-E up to epoch $T$. The results in Figure 1 in the Appendix show that our proposed estimators TSE and TSE-E using training losses clearly outperform their validation counterparts.

Another intriguing observation is that the rank correlation performance of SoVL-E drops significantly in the later phase of the training (after around 100 epochs for CIFAR10 and 150 epochs for CIFAR100)

and the final test loss, TestL (T=200), also correlates poorly with final test *accuracy*. This implies that the validation/test losses can become unreliable indicator for the validation/test accuracy on certain datasets; as training proceeds, the validation accuracy keeps improving but the validation losses could stagnate at a relatively high level or even start to rise [14, 21]. This is because while the neural network can make more correct classifications on validation points (which depend on the maximum argument of the logits) over the training epochs, it also gets more and more confident on the correctly classified training data and thus the weight norm and maximum of the logits keeps increasing. This can make the network overconfident on the misclassified *validation* data and cause the corresponding validation loss to rise, thus offsetting or even outweighing the gain due to improved prediction performance [21]. Training loss will not suffer from this problem (Appendix C). While TSE-E struggles to distinguish architectures once their training losses have converged to approximately zero, this contributes to a much smaller drop in estimation performance of TSE-E compared to that of SoVL-E and only happens near a very late phase of training (after 150 epochs) which will hardly be reached if we want efficient NAS using as *few* training epochs as possible. Therefore, the possibility of network overconfidence under misclassification is another reason for our use of training losses instead of the validation losses.

## C.2 Example showing training loss is better correlated with validation accuracy than validation loss

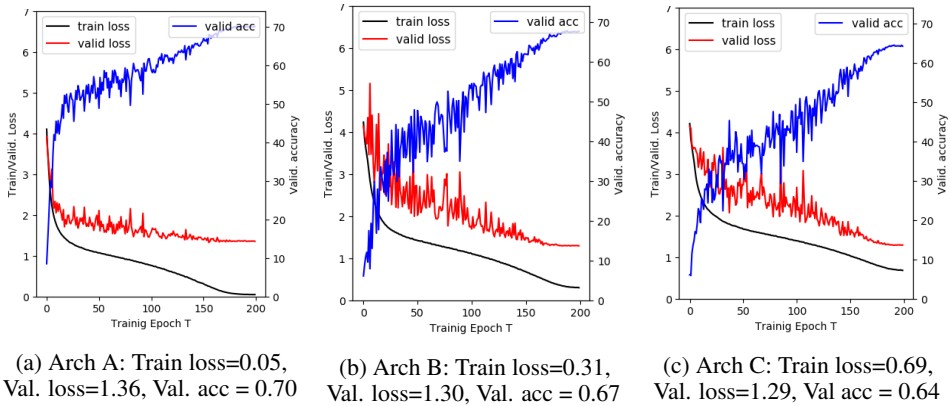

(a) Arch A: Train loss=0.05, Val. loss=1.36, Val. acc = 0.70

(b) Arch B: Train loss=0.31, Val. loss=1.30, Val. acc = 0.67

(c) Arch C: Train loss=0.69, Val. loss=1.29, Val acc = 0.64

Figure 2: Training losses, validation losses and validation accuracies of three example architectures on CIFAR100. The average of the training losses, validation losses and validation accuracies over the final 10 epochs is presented in the subcaption of each architecture.

We sample three example architectures from the NB201 dataset and plot their losses and validation accuracies on CIFAR100 over the training epochs $T$. The relative ranking for the validation accuracy is: Arch A (0.70) > Arch B (0.67) > Arch C (0.64), which corresponds perfectly (negatively) with the relatively ranking for the training loss: Arch A (0.05) < Arch B (0.31) < Arch C (0.69). Namely, the best performing architecture also has the lowest final training epoch loss. However, the ranking among their validation losses is poorly/wrongly correlated with that of validation accuracy; the worst-performing architecture has the lowest final validation losses but the best-performing architecture has the highest validation losses. Moreover, in all three examples, especially the better-performing ones, the validation loss stagnates at a relatively high value while the validation accuracy continues to rise. The training loss does not have this problem and it decreases while the validation accuracy increases. This confirms the observation we made in Appendix C.1 that the validation loss will become an unreliable predictor for the final validation accuracy as well as the generalisation performance of the architecture as the training proceeds due to overconfident misclassification. Specifically, we note that while the validation loss and training loss follow the same distribution at initialization and the train/test set inputs are assumed to be i.i.d., the train and validation losses after training are not i.i.d., particularly in models which have overfitted to their training set. This can lead to settings where a model is increasingly overconfident on some incorrect validation predictions and so obtains a large validation loss, even though its training loss and validation accuracy may be improving.

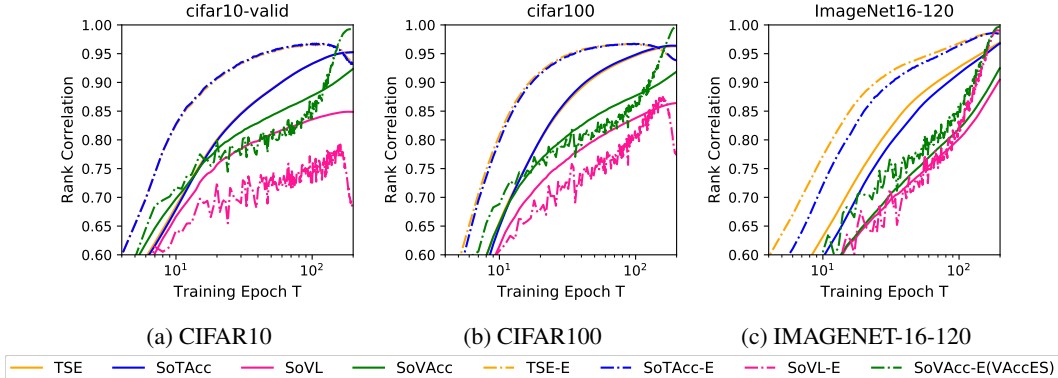

Figure 3: Rank correlation performance of our TSE (yellow), the sum over training accuracy, SoTAcc (blue), the sum over validation losses, SoVL (pink), the sum of validation accuracy, SoVAcc (green) as well as their summing over the recent $E$ epoch counterparts (dash dot) for 5000 random architectures in NB201 on three image datasets.

## C.3   Comparison with sum over accuracy

We compute another two variants of our estimator, SoTAcc and SoTAcc-E, by summing over training accuracies rather than training losses. Another two baselines to check against are the sum over validation accuracy, SoVAcc and SoVAcc-E. The results on CIFAR10 and CIFAR100 in Figure 1 of the Appendix confirm the discussion in Appendix C.1; as the training proceeds, the validation loss can become poorly correlated with the validation/test accuracy while the training loss is still perfectly correlated with the training accuracy. It is expected that SoVAcc-E should converge to a perfect rank correlation (=1) with the true test performance at the end of the training. However, the results in (a), (b) and (c) show that our proposed estimator *TSE-E can consistently outperform SoVAcc-E* in the early and middle phase of the training (roughly $T \leq 150$ epochs). This reconfirms the usefulness of our estimator.

## C.4   Overfitting on CIFAR10 and CIFAR100

In Figure 1 of Appendix C.1, the rank correlation achieved by TSE-E on CIFAR10 and CIFAR100 drops slightly after around $T = 150$ epochs but a similar trend is not observed for IMAGENET-16-120. We hypothesise that this is because many architectures converge to very small training losses on CIFAR10 and CIFAR100 in the later training phase, making it more difficult to distinguish between these good architectures based on their later-epoch training losses. However, this does not happen on IMAGENET-16-120 because it is a more challenging dataset. We test this by visualising the training loss curves of all architectures in Figure 4a of the Appendix, where the solid line and error bar correspond to the mean and standard error, respectively. We also plot out the number of architectures with training losses below 0.1 [2] in Figure 4b of the Appendix. It is evident that CIFAR10 and CIFAR100 both see an increasing number of overfitted architectures as the training proceeds whereas all architectures still have high training losses on IMAGENET-16-120 at end of the training $T = 200$ and none have overfit. Thus, our hypothesis is confirmed. In addition, similar observation is also shared in [9] where the authors find the number of optimisation iterations required to reach loss of 0.1 correlates well with generalisation but the number of iterations required to go from a loss of 0.1 to 0.01 does not.

## D   TSE Estimator Hyperparameters

Our proposed TSE estimators require very few hyperparameters: the summation window size $E$ for TSE-E and the decay rate $\gamma$ for TSE-EMA, and we show empirically that our estimators are robust to these hyperparameters. For $E$ of TSE-E, we test different summation window sizes on various search spaces and image datasets in Figure 5 and find that $E = 1$ consistently gives best

---

[2]the threshold 0.1 is chosen following the threshold for optimisation-based measures in [9]

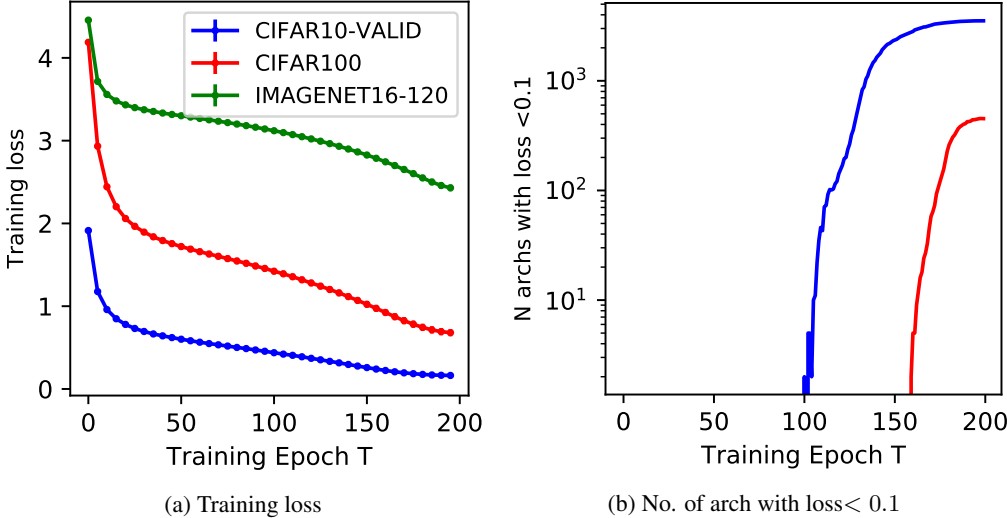

(a) Training loss

(b) No. of arch with loss$< 0.1$

Figure 4: Mean and 5 standard error of training losses and validation losses on all architectures on different NB201 image datasets. (a) shows the training curves and (b) shows the number of architectures whose training losses go below 0.1 as the training proceeds. Many architectures reach very small training loss in the later phase of the training on CIFAR10 and CIFAR100, thus may overfitting on these two datasets. But all the architectures suffer high training losses on IMAGENET-16-120, which is a much more challenging classification task, and none of them overfits.

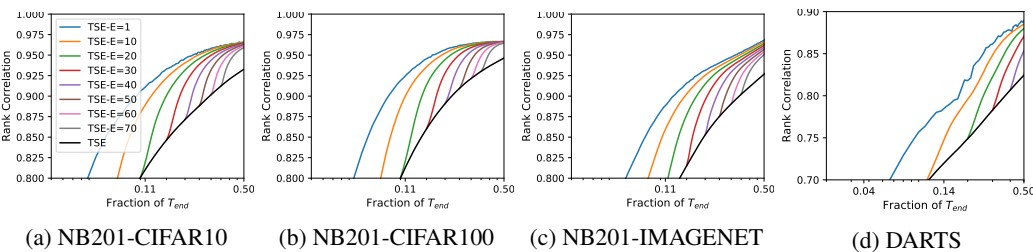

(a) NB201-CIFAR10    (b) NB201-CIFAR100    (c) NB201-IMAGENET    (d) DARTS

Figure 5: Rank correlation performance of TSE-E computed over $E$ most recent epochs. Different $E$ values are investigated for architectures in NB201 on three image datasets and 5000 architectures from NB301 (DARTS) on CIFAR10. In all cases, smaller $E$ consistently achieves better rank correlation performance with $E = 1$ being the best choice.

results across all cases. This, together with the almost monotonic improvement of our estimator's rank correlation score over the training budgets, supports our hypothesis discussed in Section **??** that training information in the more recent epochs is more valuable for performance estimation. Note that TSE-E with $E = 1$ corresponds to the sum of training losses over all the batches in one single epoch.

We also conducted an ablation study on summing training losses below one epoch, $E \in [0.1, 0.3, 0.5, 0.7]$ on the 20-cell DARTS architecture data for which we have saved each mini-batch training losses. As shown in Figure 6, again summing over the entire epoch ($E = 1$) gives the best performance; this might be because $E = 1$ covers the entire training set as Bayesian marginal likelihood and PAC-Bayes bound do, which are the theoretical inspirations for our method. On the other hand, we observe that $E \geq 0.3$ can achieve relatively close performance as $E = 1$, thus motivating our use of small number of mini-batches to estimate TSE in one-shot and gradient-based NAS experiments in the main text.

As for $\gamma$, we show in Figure 7 that TSE-EMA is robust to a range of popular choices $\gamma \in [0.9, 0.95, 0.99, 0.999]$ across various datasets and search spaces. Specifically, the performance difference among these $\gamma$ values are almost indistinguishable compared to the difference between

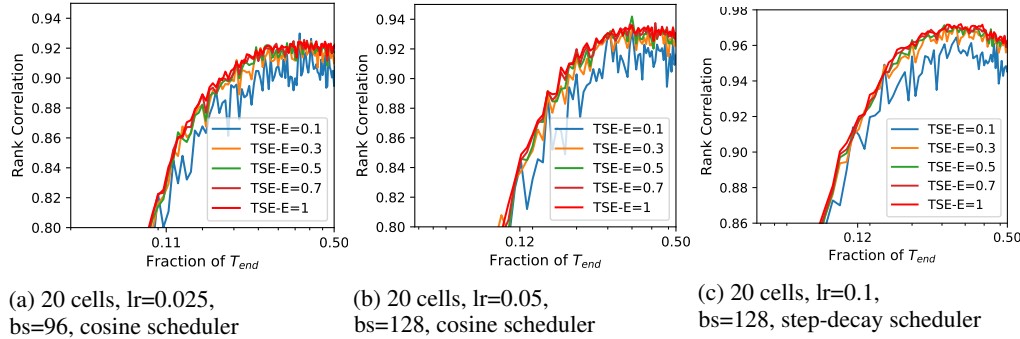

(a) 20 cells, lr=0.025,
bs=96, cosine scheduler

(b) 20 cells, lr=0.05,
bs=128, cosine scheduler

(c) 20 cells, lr=0.1,
bs=128, step-decay scheduler

Figure 6: Rank correlation performance of TSE-E computed within one epoch $E < 1$. For example, $E = 0.1$ corresponds to the sum of training losses over $10\%$ of the mini-batches in an epoch. Various $E$ values are investigated on the 20-cell DARTS architecture data which are evaluated on CIFAR10 under three different training set-ups and used for Figure 2 (b) to (d) in the main text. In all settings, $E = 1$ consistently achieves best rank correlation performance but $E \geq 0.3$ can achieve relatively close performance.

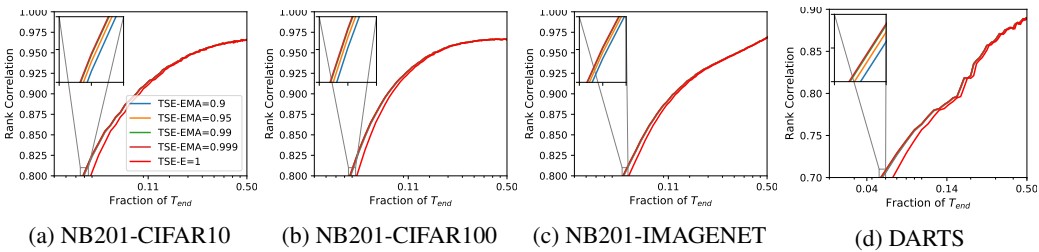

(a) NB201-CIFAR10

(b) NB201-CIFAR100

(c) NB201-IMAGENET

(d) DARTS

Figure 7: Rank correlation performance of TSE-EMA with $\gamma$ on architectures in NB201 on three image datasets and 5000 architectures from NB301 (DARTS) on CIFAR10. In all cases, TSE-EMA is very robust to different $\gamma$ values; the difference among TSE-EMA with different $\gamma$ is indistinguishable compared to that with TSE-E.

TSE-EMA and TSE-E. Thus, we set $E = 1$ and $\gamma = 0.999$ in all the following experiments and recommend them as the default choice for potential users who want to apply TSE-E and TSE-EMA on a new task without additional tuning.

# E  Architecture Generator Selection

For the RandWiredNN dataset, we use $69$ different hyperparameter values for the random graph generator which generates the randomly wired neural architecture. Here we would like to investigate whether our estimator can be used in place of the true test accuracy to select among different hyperparameter values. For each graph generator hyperparameter value, we sample $8$ neural architectures with different wiring. The mean and standard error of the true test accuracies, TSE-EMA scores and early stopped validation accuracy (VAccES) over the $8$ samples are presented in Fig. 8. Our estimator can accurately predict the relative performance ranking among different hyperparameters (Rank correlation $\geq 0.84$) and accurately identify the optimal hyperparameter (circled in black) based on as few as 10 epochs of training ($T = 10$). The prediction by VAccES is less consistent and accurate and the rank correlation between VAccES and the final test accuracy is always lower than that of our TSE-EMA across different training budgets.

# F  Effective Training Budget for Our TSE Estimators

Our estimators can achieve superior rank correlation with the true generalisation performance for a relatively wide range of training budgets $T < T_{end}$. However, our estimator is not meant to replace the validation accuracy at the end of training $T = T_{end}$ or when the user can afford large training budget to sufficient train the model. In those settings, validation accuracy remains as the gold standard

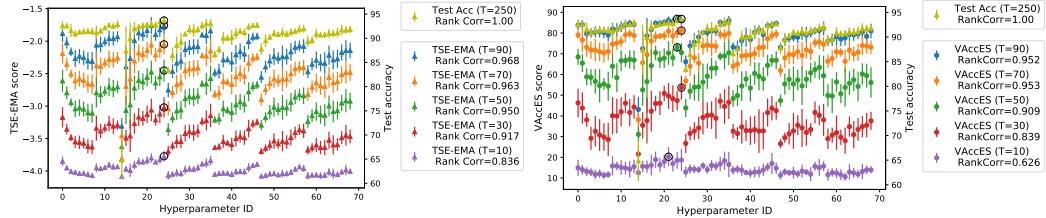

(a) Training speed estimator: TSE-EMA

(b) Early-stopped validation accuracy: VAccES

Figure 8: Model selection among 69 random graph generator hyperparamters on RandWiredNN dataset using our TSE-EMA (a) and VAccES (b). We use each hyperparameter value to generate 8 architectures and evaluate their true test accuracies after complete training. The mean and standard error of the test performance across 8 architectures for each hyperparameter value are presented as Test Acc (yellow) and treated as ground truth (Right y-axis). We then compute our TSE-EMA estimator for all the architectures by training them for $T < T_{end} = 250$ epochs. The mean and standard error of TSE-EMA scores for $T = 10, \ldots, 90$ are presented in different colours (Left y-axis of (a)). The rank correlation between the mean Test Acc and that of TSE-EMA for various $T$ is shown in the corresponding legends in (a). The same experiment is conducted by using early-stopped validation accuracy (VAccES) for performance estimation (b). With only 10 epochs of training, our TSE-EMA estimator can already capture the trend of the true test performance of different hyperparameters relatively well (Rank correlation= $0.851$) and can successfully identify 24-th hyperparamter setting as the optimal choice. The prediction of best hyperparameter by VAccES is less consistent and the rank correlation scores of VAccES at all epochs are lower than those of TSE-EMA.

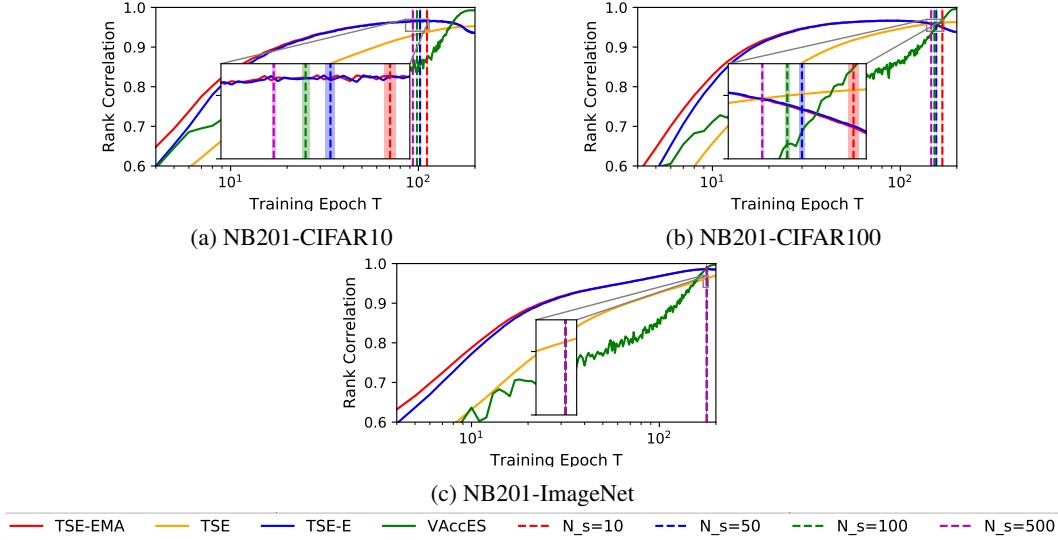

(a) NB201-CIFAR10

(b) NB201-CIFAR100

(c) NB201-ImageNet

Figure 9: Rank correlation performance up to $T = T_{end}$. If the users want to apply our estimators for large training budget, they can estimate the effective range of our estimators based on the minimum epoch $T_o$ when overfitting happens among the $N_s$ observed architectures. They can then stop our estimators early at $0.9T_o$ (marked by vertical lines) or switch back to validation accuracy beyond that.

for evaluating the true test performance of architectures. This is shown in Figure 9 that as the training budget approaches $T_{end}$, our estimators will eventually be overtaken by validation accuracy. While the region requiring large training budget is less interesting for NAS where we want to maximise the cost-saving by using performance estimators, if the user does want to apply our estimators with a relatively large training budget, we propose a simple method here to estimate when our estimators would be less effective than validation accuracy.

As discussed in Appendix C.4, we notice that that our estimators, TSE-EMA and TSE-E, become less effective when the architectures compared start to overfit because both of them rely heavily on the lastest-epoch training losses to measure training speed, which is difficult to estimate when the

---

**Algorithm 1** Find Effective Training Budget for TSE Estimators

---

1: **Input:** A subset of $N_s$ fully trained architectures whose training losses are $\{\{\ell_{i,t}\}_{t=1}^{T_{end}}\}_{i=1}^{N_s}$,
   Overfitting criterion $is\_overfit()$
2: **Output:** The effective training budget $T_{effective}$ to use TSE-EMA or TSE-E
3: $\mathcal{S}_{T_o} = \emptyset$
4: **for** $i = 1, \ldots, N_s$ **do**
5:    **for** $t = 1, \ldots, T_{end}$ **do**
6:       **if** $is\_overfit(\ell_{i,t}) ==$ True **then**
7:          $T_{i,o} = T$
8:          break
9:       **else**
10:          $T_{i,o} = T_{end}$
11:       **end if**
12:    **end for**
13:    $\mathcal{S}_{T_o} = \mathcal{S}_{T_o} \cup T_{i,o}$
14: **end for**
15: $T_o = \min \mathcal{S}_{T_o}$
16: $T_{effective} = 0.9 T_o$

---

training losses become too small. Thus, we can adopt the heuristic described in Algorithm 1 to decide when to stop the computation of TSE-E and TSE-EMA early and revert to a previous checkpoint i.e. $T_{effective}$ [3]. Similar to Appendix C.4, we use a simple criterion: training loss decreasing below 0.1 (i.e. $\ell_{i,t=T_o} < 0.1$), to identify when an architecture start to overfit. We experiment with $N_s = 10, 50, 100, 500$ architectures and the mean and standard error results over 100 random seeds are shown in Figure 9. It's evident that we can find a fairly reliable threshold with a sample size as small as $N_s = 10$.

## G  Additional NAS experiments

### G.1  Query-based NAS

In this work, we incorporate our estimator, TSE-EMA, at $T = 10$ into three NAS search strategies: Regularised Evolution [18], Bayesian optimisation [2] and Random Search [1] and performance architecture search on NB201 datasets. We modify the implementation available at `https://github.com/automl/nas_benchmarks` for these three methods.

Random Search [1] is a very simple yet competitive NAS search strategy [7]. We also combined our estimator, TSE-EMA, at training epoch $T = 10$ with Random Search to perform NAS. We compare it against the baselines using the final validation accuracy at $T = 200$, denoted as Val Acc (T=200), and the early-stop validation accuracy at $T = 10$, denoted as Val Acc (T=10). Other experimental set-ups follow Section 4.3. The results over running hours on all three image tasks are shown in Figure 10 of the Appendix. Note the x-axis is in log scale. The use of our estimator clearly leads to faster convergence as compared to the use of final validation i.e. Val Acc (T=200). Moreover, our estimator also slightly outperforms the early-stop validation accuracy, Val Acc (T=10) on the three image tasks. The performance gain of using our estimator or the early-stopped validation accuracy is relatively less significant in the case of Random Search compared to the cases of Regularised Evolution and TPE. For example, given a budget of 50 hours on CIFAR100, Regularised Evolution and TPE when combined with our estimator can find an architecture with a test error around or below 0.26 but Random Search only finds architecture with test error of around 0.268. This is due to the fact that Random Search is purely explorative while Regularised Evolution and TPE both trade off exploration and exploitation during their search; our estimator by efficiently estimating the final generalisation performance of the architectures will enable better exploitation. Therefore, we recommend the users to deploy our proposed estimator onto search strategies which involve some degree of exploitation to maximise the potential gain.

---

[3] We use $T = 0.9 T_o$ rather than $T_o$ to be conservative.

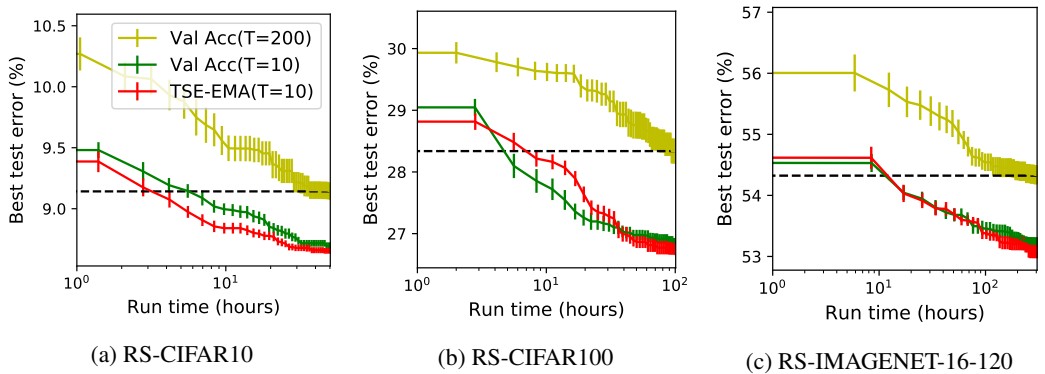

| (a) RS-CIFAR10 | (b) RS-CIFAR100 | (c) RS-IMAGENET-16-120 |

Figure 10: NAS performance of Random Search (RS) in combined with final validation accuracy (Final Val Acc), early-stop validation accuracy (ES Val Acc) and our estimator TSE-EMA on NB201. TSE-EMA enjoys competitive convergence as ES Val Acc and both are faster than using Final Val Acc.

## G.2 One-shot NAS

We follow the RandNAS[11] procedure for the supernetwork training but modify the search phase: for each randomly sampled subnetwork, we train it for $B$ additional mini-batches after inheriting weights from the trained supernetwork to compute our TSE estimator. Note although this introduces some costs, our estimator saves all the costs from evaluation on validation set as it doesn't require validation data. To ensure fair comparison, we also recompute the validation accuracy, SoVL, Tlmini of each subnetwork after the additional training. We also experiment with more advanced supernetwork training techniques such as FairNAS [6] and MultiPaths [27] and show that our estimators can be applied on top of them to further improve the rank correlation performance. Please refer to Section 4.4 for more experimental details and to the start of Section 4 for description of different estimators. The complete table of results with more estimators are presented in Table 2 in Appendix.

Table 2: Results of performance estimators in one-shot NAS setting over 3 supernetwork training initialisations. For each supernetwork, we randomly sample 500 random subnetworks for DARTS and 200 for NB201, and compute their TSE, Val Acc, SoVL, Tlmini after inheriting the supernetwork weights and training for $B$ additional minibatches. Rank correlation measures the estimators' correlation with the rankings of the true test accuracies of subnetworks when *trained from scratch independently*, and we compute the average test accuracy of the top 10 architectures identified by different estimators from all the randomly sampled subnetworks.

| B | Estimator | Rank Correlation | | | | Average Accuracy of Top 10 Architectures | | | |
| | | NB201-CIFAR10 | | | DARTS | NB201-CIFAR10 | | | DARTS |
| | | RandNAS | FairNAS | MultiPaths | RandNAS | RandNAS | FairNAS | MultiPaths | RandNAS |
| 100 | TSE | **0.70 (0.02)** | **0.84 (0.01)** | **0.83 (0.01)** | **0.30(0.04)** | **92.67 (0.12** | **92.70 (0.10)** | 92.63 (0.12) | **93.64(0.04)** |
| | Val Acc | 0.44 (0.15) | 0.56 (0.17) | 0.67 (0.05) | 0.11(0.04) | 91.47 (0.31) | 91.73 (0.21) | 91.77 (0.78) | 93.20(0.04) |
| | SoVL | 0.54 (0.13) | **0.84 (0.06)** | **0.83 (0.01)** | 0.10(0.04) | 92.57 (0.15) | 92.67 (0.06) | **92.73 (0.06)** | 93.21(0.04) |
| | Tlmini | 0.62 (0.03) | 0.72 (0.09) | 0.74 (0.02) | 0.05(0.03) | 91.80 (0.40) | 92.33 (0.40) | 92.43 (0.06) | 93.38(0.03) |
| 200 | TSE | **0.70 (0.03)** | **0.850 (0.01)** | **0.83 (0.01)** | **0.32(0.04)** | **92.70 (0.00)** | **92.77 (0.06)** | **92.73 (0.06)** | **93.55(0.04)** |
| | Val Acc | 0.41 (0.10) | 0.56 (0.17) | 0.53 (0.11) | 0.09(0.02) | 91.53 (0.55) | 92.40 (0.10) | 92.23 (0.23) | 93.34(0.02) |
| | SoVL | 0.52 (0.17) | 0.84 (0.06) | 0.80 (0.02) | 0.08(0.02) | 90.70 (1.35) | 92.53 (0.15) | 92.50 (0.10) | 93.36(0.02) |
| | Tlmini | 0.46 (0.14) | 0.72 (0.10) | 0.69 (0.06) | 0.02(0.01) | 92.00 (0.35) | 92.53 (0.25) | 92.40 (0.27) | 93.15(0.01) |
| 300 | TSE | **0.71 (0.03)** | **0.85 (0.00)** | **0.82 (0.01)** | **0.34(0.04)** | **92.70 (0.00)** | **92.77 (0.06)** | **92.70 (0.00)** | **93.65(0.04)** |
| | Val Acc | 0.44 (0.04) | 0.62 (0.08) | 0.59 (0.71) | 0.06(0.02) | 91.20 (0.35) | 92.10 (0.50) | 91.43 (0.72) | 93.31(0.02) |
| | SoVL | 0.45 (0.21) | 0.81 (0.05) | 0.81 (0.03) | 0.05(0.02) | 91.00 (1.60) | 92.53 (0.15) | 92.53 (0.06) | 93.26(0.02) |
| | Tlmini | 0.47 (0.12) | 0.74 (0.02) | 0.70 (0.04) | 0.09(0.01) | 91.60 (0.44) | 92.43 (0.21) | 92.27 (0.12) | 92.95(0.01) |

## G.3 Differentiable NAS

We modify two differentiable approach, DARTS [12] and DrNAS [4], by directly using the derivative of our TSE estimator instead of that of the original validation loss to update the architecture (distribution) parameters.

---

**Algorithm 2** DARTS

---

1: Create a mixed operation $\bar{o}^{i,j}$ parametrised by $\alpha^{i,j}$ for each edge $(i,j)$
2: **for** $t = 1, \ldots, BT$ **do**
3:     Update architecture parameter $\alpha$ by descending $\nabla_\alpha \ell_{val}(w, \alpha)$
4:     Update weights $w$ by descending $\nabla_w \ell_{train}(w, \alpha)$
5: **end for**
6: Derive the final architecture based on the learned $\alpha$

---

**Algorithm 3** DARTS-TSE

---

1: Create a mixed operation $\bar{o}^{i,j}$ parametrised by $\alpha^{i,j}$ for each edge $(i,j)$
2: **for** $t = 1, \ldots, \lfloor BT/K \rfloor$ **do**
3:     Update architecture parameter $\alpha$ by descending $\nabla_\alpha \ell_{TSE}(w, \alpha)$
4:     $\nabla_\alpha \ell_{TSE}(w, \alpha) = 0$
5:     **for** $k = 1, \ldots, K$ **do**
6:         Update weights $w$ by descending $\nabla_w \ell_{train}(w, \alpha)$
7:         $\nabla_\alpha \ell_{TSE}(w, \alpha) = \nabla_\alpha \ell_{TSE}(w, \alpha) + \nabla_w \ell_{train}(w, \alpha)$
8:     **end for**
9: **end for**
10: Derive the final architecture based on the learned $\alpha$

---

In DARTS [12], each intermediate node $\phi^{(j)}$ is computed based on all of its predecessors:

$$\phi^{(j)} = \sum_{i<j} \bar{o}^{(i,j)}\left(\phi^{(i)}\right) \tag{6}$$

where $\bar{o}^{(i,j)}(\phi)$ is a mix of all possible operations $o(\phi)$:

$$\bar{o}^{(i,j)}(\phi) = \sum_{o \in \mathcal{O}} \frac{\exp(\alpha_o^{(i,j)})}{\sum_{o' \in \mathcal{O}} \exp(\alpha_{o'}^{(i,j)})} o(\phi) \tag{7}$$

The architecture parameters to be searched in DARTS is thus a set of continuous vectors $\alpha = \{\alpha^{(i,j)}\}$. Assume we run the search for $T$ epochs and each epoch comprises $B$ mini-batches, the algorithms of DARTS and DARTS-TSE is summarised in Algorithm 2 and 3. Note in DARTS, the architecture parameters are updated with the derivative of validation loss $\nabla_\alpha \ell_{val}(w, \alpha)$ at each mini-batch, leading to a total of $BT$ updates. In DARTS-TSE, we compute the TSE estimator and its derivative using $K = 100$ mini-batches, leading to a less frequent update of $\alpha$. To compensate that, we set $\nabla_\alpha \ell_{TSE}(w, \alpha) = \sum_{k=1}^{K} \nabla_w \ell_{train}^{(k)}(w, \alpha)$ instead of $\nabla_\alpha \ell_{TSE}(w, \alpha) = \frac{1}{K} \sum_{k=1}^{K} \nabla_w \ell_{train}^{(k)}(w, \alpha)$ for updating $\alpha$.

DrNAS [4] is very similar to DARTS but instead of updating the architecture parameters $\alpha$ directly, DrNAS assumes $\alpha$ is drawn from a Dirichlet distribution $q(\alpha|\beta)$ and optimise the distribution parameters $\beta$. $\beta$ is updated at each mini-batch by descending:

$$\mathbb{E}_{q(\alpha|\beta)}\left[\nabla_\beta \ell_{val}(w, \alpha)\right] \tag{8}$$

In DrNAS-TSE, we instead use the derivative of TSE to update $\beta$:

$$\sum_{k=1}^{B} \mathbb{E}_{q(\alpha|\beta)}\left[\nabla_\beta \ell_{train}^k(w, \alpha)\right] \tag{9}$$

For both DARTS and DrNAS, we set $K = 100$ for computing our TSE and follow the default setting in [7, 12, 4] for all the other hyperparameters including $B$ and $T$.

### G.4   NAS results with the DARTS training protocol

We re-evaluate the DARTS architectures found by RandNAS-TSE and DrNAS-TSE following the DARTS training protocol [12] (i.e. 20-cells, 600 epochs, auxiliary towering and cut-off) on CIFAR10.

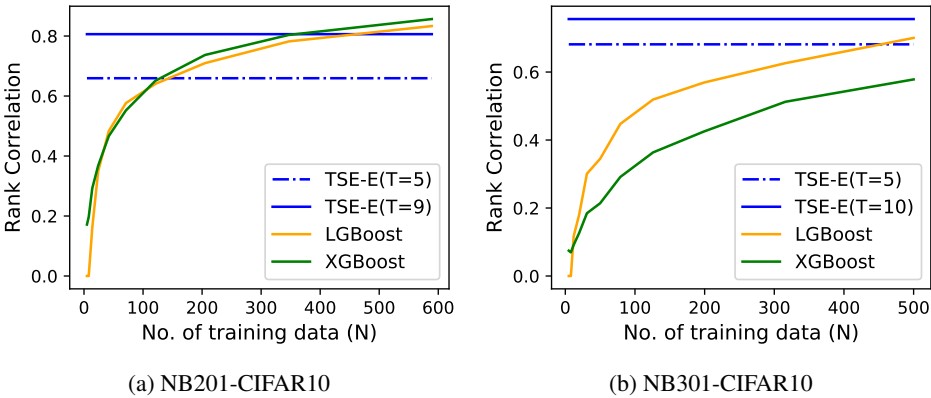

|                | (a) NB201-CIFAR10 | (b) NB301-CIFAR10 |
|----------------|-------------------|-------------------|

Figure 11: Rank correlation performance of TSE-E computed with $T$ epochs of training as well as LGBoost and XGBoost with $N$ training data. For example, TSE-E(T=10) means TSE-E score computed at training epoch 10.

For fair comparison, we also run RandNAS and DrNAS on DARTS space using their open-sourced codes and fully re-evaluate the found optimal architecture cells. The default seed recommended in the open-sourced codes are used for both RandNAS and DrNAS as well as their TSE variants. The results in Table 3 show that the use of our TSE can lead to improvement after full training protocol.

Regarding DrNAS, we are also aware of the difference between our results and those in the original DrNAS paper. In fact, we tried our best to reproduce DrNAS results by using their open-sourced code and following the default settings as well as seed in the code. However, we still fail to reproduce the results reported in their paper for any of the NB201 results as well as for the DARTS search space on CIFAR10. We admit there might be some stochasticity in the search process and we would contact the authors on this issue. Although we also notice that there is a similar issue raised in the code repository of DrNAS, complaining about the inability to reproduce DrNAS's CIFAR10 results using even the architecture (genotype) recommended in the code.

Table 3: Comparison among the optimal DARTS architectures on CIFAR-10 found by different NAS strategies and retrained following the full training protocol [12].

| NAS Method  | Test Accuracy (%) |
|-------------|-------------------|
| RandNAS     | 96.87             |
| RandNAS-TSE | **97.21**         |
| DrNAS       | 96.90             |
| DrNAS-TSE   | **97.05**         |

## H    Additional Ablation Studies

### H.1    Comparison to other model-based predictors

Recent works [20, 23] have shown that alternative model-based estimators such as LGBoost [10] and XGBoost [3] can achieve superior performance in predicting the architecture test performance. We compared our TSE methods to them on NB201-CIFAR10 and NB301-CIFAR10. To ensure fair comparison, We tried different numbers of training data ($N$) for LGBoost and XGBoost (e.g. $N = 200$ means 200 architecture data for training) following [23] and optimised their hyperparameters for each $N$ by running random search in the ranges proposed in [20] for 1000 iterations. The Spearman's rank correlation results are shown in Figure 11. Note TSE-E(T=10) means TSE-E score computed at training epoch 10. It's evident that our TSE estimators with only $T = 5$ epochs can already outperform LGBoost and XGBoost with $N = 200$ and TSEs with $T = 10$ epochs can outperform LGBoost and XGBoost with $N = 500$. Note the cost for collecting such a large amount of architecture data (e.g. $N = 500$) to fit the prediction model can be very expensive. Interestingly, a

concurrent work [23] compares our TSE estimators (named as SoTL-E and SoTL in [23]) against more model-based predictors and shows that our TSE can outperform most of them with small amount of training budget.

## H.2  Does having more skip-connections necessarily lead to faster convergence?

Interestingly, we find that the use of our TSE estimator in place of validation losses can actually prevent DARTS from overfitting to the skip connections: the final architecture cells found by DARTS in Figure 5 (b) of the main text contains 6 skip connections but those by DARTS-TSE contains only 1 skip connection.

Also the claim in [28] that "the more skip connections the faster convergence" are not extensively verified with empirical evidence on NAS architectures. For example on NB201, we limited the operation choices to {skip_connect, conv3x3} and checked the mean test performance (Test Acc) as well as mean TSE-E scores at training epoch $T = 10$ and $T = 50$, denoted as TSE-E(T=10) and TSE-E(T=50), for *all* cells containing the corresponding number of skip connections ($N_{skip}$). The results in Table 4 show that more skip connections don't necessarily lead to faster training speed as measured by TSE (smaller TSE means faster training speed) even at the very early training phase of epoch $T = 10$ and TSE-E(T=10) can already predict almost perfectly the generalisation performance of architectures with different number of skip connections. So the results in Table 4 dispute that architectures with more parameter-free operations necessarily train faster and our experiments support our claim that architectures that trains faster still lead to better generalisation performance in general.

Table 4: The average test accuracy and TSE-E scores over architecture cells containing $N_{skip}$ skip connections. Note TSE-E(T=10) denotes TSE-E score computed with 10 epochs of training.

| $N_{skip}$ | Test Accuracy (%) | TSE-E(T=10) | TSE-E (T=50) |
|---|---|---|---|
| 1 | 89.87 | 0.74 | 0.32 |
| 2 | 90.01 | 0.59 | 0.29 |
| 3 | 89.45 | 0.65 | 0.32 |
| 4 | 86.48 | 0.92 | 0.54 |
| 5 | 79.82 | 1.18 | 0.86 |
| 6 | 39.98 | 1.80 | 1.71 |

# I  Broader Impact

**Making NAS more environmentally friendly**   Training a deep neural network can lead to a fair amount of carbon emissions [22]. Such environmental costs are significantly amplified if we need to perform NAS [22] where repeated training is resource-wasteful but necessary. Our work proposes a cheap yet reliable alternative for estimating the generalisation performance of a neural network based on its early training losses; this significantly reduces the training time required during NAS and thus decreases the corresponding environmental costs incurred. Note although developed for the NAS setting, our estimator are potentially applicable for model selection in general as demonstrated in Appendix E, both of which is frequently performed by almost all machine learning practitioners. While our estimator can hardly be on par with the fully trained test accuracy in assessing the generalisation performance of a model, if the practitioners could adopt our estimator in place of looking at the fully trained test accuracy as often as possible, the environmental cost-saving would be substantial.

**Making NAS accessible to more users and accelerating the development of NAS search strategies**   By speeding up the NAS performance evaluation, our work can reduce not only the computational resources required to run many current NAS search strategies but also the sunk costs incurred during the process of developing new search strategies. This increases the chance of the researchers or users, who have limited computing budgets, being able to use or study NAS, which may in turn stimulate the advancement of NAS research. On a broader scale, this also helps NAS better serve its original motivation which is to free human-labour from designing neural networks for new tasks and make good machine learning models easily accessible to general community. However, a potential negative impact of this is that with more practitioners using NAS and thus deep learning,

the overall increase (due to increase in usage) in the environmental costs associated might outweigh the resource-saving (for each usage) mentioned above.