# OpenReview forum: "Speedy Performance Estimation for Neural Architecture Search"
_NeurIPS.cc/2021/Conference — NeurIPS 2021 Spotlight_

### Official Review · Reviewer_BhHR · 2021-07-07

**Rating:** 7
**Confidence:** 4

**Summary:**

This paper proposes a simple yet effective model-free performance evaluation metric for architectures with partial training. To briefly summarize, the proposed method uses the (weighted) average of training loss over multiple mini-batches. Intuitively speaking, this metric measures the variance of gradients across mini-batches to some extent, which is related to the ease of convergence. Extensive empirical evaluations are conducted across different spaces and the results show that the proposed metric consistently outperforms existing ones; The theoretical aspect of this paper includes a connection of TSE to PAC bound. This work has its value as an add-on to many existing NAS algorithms. For these reasons, I vote for acceptance. I do have a few questions for the author, which I will explain in the detailed comment. And I might adjust my opinion of the paper depending on the responses.

**Limitations And Societal Impact:**

Yes, it is addressed in the appendix.

**Main Review:**

Strength:
1. The author demonstrates that the proposed method can be used as a simple add-on to improve many existing NAS methods.
2. Solid experiments, the author studies on a wide range of spaces (e.g. NB201, NB301, DARTS, Randomly wired spaces, ResNeXT space, e.t.c.) and demonstrated consistent results.
3. ranking among the top 1% architectures on some spaces is also reported, which is an important measure of the effectiveness of the proposed method, especially for NAS.
4. sufficient discussion/experiments are provided for the hyperparameter of the proposed metric (i.e. window size and decay ratio). A procedure to decide the training budget is also provided.

Weakness:
1. It seems that the author tests the ranking correlation on larger scale search spaces like DARTS Space, but does not report and compare the final results. Perhaps turning the proposed metric into a full search algorithm (by combining it with random sampling or some other existing method) and evaluating on DARTS Space might make the paper stronger. Like those experiments in Figure 5. But this is not a major point.
2. Please refer to the questions in the Detailed Comment Section.

Detailed Comments:
1. The paper mentioned that the training speed is related to generalization. However, in NAS, people generally believe that architectures that are easier to train (e.g. with more parameter-free operations) might stand out in the beginning but plateau quickly and therefore falls behind eventually [2]. These two conclusions seem a bit contradictory at first sight. I would like to hear the author's comment on this matter.

2. It is interesting to see that using training loss works better than the valid loss in TSE. I wonder if the same result can be observed in early stopping: Could you also show the curves of early stopping + training loss and early stopping + valid loss in Figure 1 (a-c would be sufficient)?

3. For DrNAS in Figure 5a: this curve seems different from their original paper [1] (Figure 1). From your curve, it seems that the performance of DrNAS drops to ~50% on 201 (which is basically an architecture full of skip/none) in the middle run. But in Figure 1 of the original paper, the solid line - the test accuracy of the best architecture derived by argmax of the concentration parameter - seems pretty stable. Do you have any idea about the difference?

Minors:
1. You might want to move Figure 1 up to page 3/4, as it is cued twice there.

[1] Chen et al. DrNAS: Dirichlet Neural Architecture Search (ICLR2021)\
[2] Zhou et al. Theory-Inspired Path-Regularized Differential Network Architecture Search (NIPS2020)

**Time Spent Reviewing:**

3.5

---

> ### Author Response · Authors · 2021-08-10
> **Thanks for your detailed comments and our responses are below**
>
> 1.	>*Evaluate on DARTS space*
>
> Following your suggestion, please refer to our response 2 to Reviewer Y2bt where we provide the requested experiments.
>
> 2.	>*Failure to reproduce the results in the original DrNAS paper*
>
> We are also aware of the difference between our results in Fig. 5 and that in the original DrNAS paper. In fact, we tried our best to reproduce DrNAS results by using their open-sourced code and following the default settings/seeds in the code. However, we still couldn’t reproduce the results reported in their paper for any of the NAS-Bench-201 results as well as for the DARTS search space on CIFAR10. As we mentioned in our response 2 to Reviewer Y2bt above,  we got an architecture with test error = 3.1% by running the code of DrNAS for searching and re-training. We admit there might be some stochasticity in the search process but the difference seems too big to be caused by stochasticity alone. We would contact the authors on this issue, although we also notice that there is a similar issue raised in the code repository of DrNAS, complaining about the inability to reproduce DrNAS’s CIFAR10 results using even the architecture (genotype) recommended in the code.
>
> 3.	>*Not a major point… architectures that are easier to train might converge faster in the beginning but falls behind eventually (Zhou et al., 2020)*
>
> We are also aware of (Zhou et al., 2020) and interestingly, we find that the use of our TSE estimator in place of validation losses can actually prevent DARTS from overfitting to the skip connections:  the final architecture cells found by DARTS in Fig. 5 (b) contains 6 skip connections but those by DARTS-TSE contains only 1 skip connection. Also the claim in (Zhou et al., 2020) that “the more skip connections the faster convergence” are not extensively verified with empirical evidence on NAS architectures. For example on NAS-Bench-201, we limited the operation choices to {skip_connect, conv3x3} and checked the mean test performance (Test Acc) as well as mean TSE-E scores at training epoch T=10 and T=50 for all cells containing different number of skip connections (N_skip). The results below show that more skip connections don’t necessarily lead to faster training speed as measured by TSE (smaller TSE means faster training speed) even at the very early training phase of epoch T=10 and TSE-E(T=10) can already predict almost perfectly the generalisation performance of architectures with different number of skip connections.
>
> | N_skip | Test Acc (%) | TSE-E(T=10) Score | TSE-E(T=50) Score |
> | :---: | :---: | :---: | :---: |
> | 1 | 89.87 | 0.74 | 0.32 |
> | 2 | 90.01 | 0.59 | 0.29 |
> | 3 | 89.45 | 0.65 | 0.32 |
> | 4 | 86.48 | 0.92 | 0.54 |
> | 5 | 79.82 | 1.18 | 0.86 |
> | 6 | 39.98 | 1.80 | 1.71 |
> | | | | |
>
> So the above results dispute that architectures with more parameter-free operations necessarily train faster and our experiments support our claim that architectures that trains faster still lead to better generalisation performance in general.
>
> 4.	>*If the same result can be observed in early stopping: Could you also show the curves of early stopping + training loss and early stopping + valid loss in Figure 1 (a-c would be sufficient)*
>
> We are sorry that we didn’t quite get this comment. Would you mind further clarifying on this point in the discussion phase?
>
> 5.	>*Move Figure 1 up to page 3/4*
>
> Thanks for the suggestion on the formatting.

---

> > ### Comment · Reviewer_BhHR · 2021-08-25
> > **Thank you for the reply**
> >
> > I apologize for the delay in my responses.
> > 1) The author's responses are informative (In particular the answer to Comment 1) and addressed my questions.
> > 2) To my knowledge DrNAS has two versions (w and w/o progressive learning). My guess is that you probably used the second version. My understanding is that their progressive learning is meant to trade accuracy for memory efficiency for large search spaces, so it is not necessary for small spaces like 201. Nevertheless, the extensive study provided by the paper elsewhere makes it a minor point, and therefore would not alter my opinion.
> >
> > This is a solid work.

---

### Official Review · Reviewer_Y2bt · 2021-07-10

**Rating:** 7
**Confidence:** 5

**Summary:**

This work proposes that the summing of training losses may be a better indicator than validation accuracy especially at the early phase of architecture search. Author[s] provides experimental results on various model-free speedy estimators, zero-cost estimators, model-based extrapolators; and shows that TSE-E and TSE-EMA are the most effective two among them.

**Limitations And Societal Impact:**

This is solid work. Experiments have been conducted in a comprehensive way in my opinion. In addition to the improvement suggestions as pointed above, there is not too much I can think of now. If time is allowed, author[s] may provide experimental results compared to SOTA NAS results on DARTS search space with the same training protocol as used in DARTS original code. In Line 226, LcSVR (or in general, model-based predictor) can be optimized in an active learning way with the updated pool instead of using a large initial population for a one-time fit to save the computational resources, as used in many recent works such as MLPs in BANANAS, GCN with BLR in BONAS, and DNGO in arch2vec and CATE. This work does not contain any potential negative societal impact.

**Main Review:**

Merits:
1. Good motivation. The paper starts with the generalization bound based on the properties of the training trajectory, hypothesizing that training speed may be a less noisy measure of generalization compared to the validation accuracy in the view of NAS, where the search budget is usually limited.

2. Solid experiments. The paper conducted experiments on NAS201, RandWire, NDS, DARTS to evaluate the ranking correlation preserving the capability of these performance estimators, as well as on NAS201,  NAS301 (DARTS) to evaluate the NAS performance of both query-based (where they evaluate three subroutines RS, RE, BO) and weight-sharing (where they evaluate four subroutines RandNAS, FairNAS, MultiPahs, Bi-level). In all these experiments, they show that TSE-E and TSE-EMA can better preserve the ranking correlation than SoVL and VAccES; and much better than zero-cost estimators when training progresses (though it is a little bit unfair since zero-cost is designed for epoch 0, depending on how many budgets you have).  They also show its speed-up performance under query-based NAS as well as better search performance under weight-sharing NAS.

3. Simple and effective. Figure 5 in Appendix C.4 looks surprisingly consistent across different datasets and search spaces in terms of rank correlation performance of TSE-E computed over E most recent epochs. If it holds true, in practice it means we can simply query the training loss API in existing NAS benchmarks at each epoch and directly use it as the feedback signal.

4. Code is provided.


Improvement suggestions:
1. In Line 98, you mentioned measuring training speed solely based on the epochs is likely to suffer overfitting on training data thus cause degraded model selection performance. However, in the experiments the minimum T you choose is 1, meaning that at least all mini-batches within a most recent epoch are used to compute the sum of training losses. I admit that currently NAS benchmarks only provide epoch-level statistics, which may restrict the experimental setup. Nevertheless, could you provide some discussion/results on TSE results based on mini-batch beyond discretized epochs? e.g. conduct some ablations on the actual search space instead of benchmarks so you can run TSE within an epoch.

2. Figure 2 in Appendix C.2 validates the motivation by showing that validation loss may negatively correlate with validation accuracy as training progresses. I don't quite understand why the discrepancy between validation loss and training loss is very large on CIFAR-100 since it is assumed to be i.i.d.

3. I would suggest author[s] take a look at another line of speed-up NAS in the related work. e.g. https://arxiv.org/pdf/2006.06936.pdf, https://arxiv.org/pdf/2102.07108.pdf, https://arxiv.org/pdf/2007.04965.pdf, https://arxiv.org/pdf/2105.06369.pdf. The idea is to design a better neighborhood-aware encoding/representation to build a flatter performance landscape and improve downstream subroutines in the scope of both query-based and weight-sharing based. In practice, architecture parameters could be further combined with partial learning curves to boost the KT and NAS performance, see this for a take https://cvpr21-nas.com/resources/upload/0894f613d1c1/file/1624351763247.pdf.

**Time Spent Reviewing:**

more than 12 hours

---

> ### Author Response · Authors · 2021-08-10
> **Thanks for your valuable suggestions and our responses are below**
>
> 1.	>*Ablation on TSE computed within an epoch*
>
> Thanks for your suggestion. Please refer to our response 1 to Reviewer Ctey where we provide the requested experiments.
>
> 2.	>*NAS results obtained with the DARTS training protocol*
>
> Following the reviewers’ suggestion, we re-evaluate the DARTS architectures found by RandNAS-TSE and DrNAS-TSE following the DARTS training protocol (20-cells, 600 epochs, auxiliary towering etc). For fair comparison, we also run RandNAS and DrNAS on DARTS space using their open-sourced codes and fully re-evaluate the found optimal architecture cells. The default seed recommended in the open-sourced codes are used for both RandNAS and DrNAS as well as their TSE variants. The results below show that the use of our TSE can lead to improvement after full training protocol.
>
> - RandNAS-TSE (test error = **2.79%**) < RandNAS (test error = 3.13%)
> - DrNAS-TSE (test error = **2.95%**) < DrNAS (test error = 3.10%)
>
> This is consistent with our findings in Sec. 4.4. Please refer to response 2 to Reviewer BhHR for why we fail to reproduce the DrNAS result reported in their paper.
>
> 3.	>*Why the discrepancy between validation loss and training loss is very large on CIFAR-100 since it is assumed to be i.i.d.*
>
> We note that while the validation loss and training loss follow the same distribution at initialization and the train/test set inputs are assumed to be i.i.d., the train and validation losses after training are not i.i.d., particularly in models which have overfit to their training set. This can lead to settings where a model is increasingly overconfident on some incorrect validation predictions and so obtains a large validation loss, even though its training loss and validation accuracy may be improving.
>
> 4.	>*Additional references on speed-up NAS*
>
> Thanks for the suggested references. We would include them in the related work section.

---

> > ### Comment · Reviewer_Y2bt · 2021-08-22
> > **Response**
> >
> > Thanks for the response. It sounds nice to me. Please consider including them into the final version.

---

> > > ### Author Response · Authors · 2021-08-25
> > > **Thanks for your response**
> > >
> > > We are glad that our responses have addressed your comments and we will definitely include the additional results, explanations and references into our final version. Many thanks again for your helpful comments and suggestions on improving our work.

---

### Official Review · Reviewer_2hFh · 2021-07-16

**Rating:** 8
**Confidence:** 4

**Summary:**

This paper proposes a simple yet reliable estimator for fully trained network performance inspired by the PAC-Bayes bounds. Based on the observation in previous work that networks that train faster have better generalization performance, they propose that small areas under the training loss curve early in training correlate with better final performance. They propose several variations of this estimator with varying methods of weighting more recent epochs more than early epochs. They demonstrate the effectiveness of their estimators to provide good ranking correlation performance with the final accuracy with limited training on several search spaces/datasets such as NB-201-cifar10/cifar100/imagenet16 and Resnet/Resnext-cifar10. They also demonstrate that their estimator provides better rankings and find better performing networks compared to using the validation accuracy when applied to the problem of selecting a final architecture from a trained super-network for one-shot NAS. They also demonstrate that it can more efficiently find high performing networks compared to validation accuracy when applied to Regularized Evolution NAS and Bayesian Optimization NAS.

**Limitations And Societal Impact:**

I believe that limitations and societal impacts are sufficiently addressed.

**Main Review:**

This paper tackles the problem of finding estimators that can better predict final network performances with few epochs of training. Their proposed estimator based on the area under the training curve is simple and very efficient to calculate and they demonstrate that it is robust with thorough experiments with multiple use cases, multiple search spaces, and multiple datasets. I strongly recommend this paper to be accepted. While the method is not incredibly novel and there is previous work on early stopping, their paper has significance due to the simplicity of the method and the range of experimental evidence they provide to demonstrate their algorithm

Strengths:
A large strength of this paper is in its simplicity. It target's an important area of model-free accuracy prediction from a single training run. Performance predictors are a very important field of work for NAS and this method has strong performance utilizing only the training loss compared to weight-sharing proxies and model-based performance estimators which requires you start with a relatively large number of trained models.

The submission is well supported by their diverse experimental results. They demonstrate their simple metric works well to rank architectures on CIFAR10, CIFAR100, Imagenet16, Flower102. They show that it can speed up Regularized Evolution, Bayesian Optimization, and one-shot NAS. They also show that it can rank architectures well in the NB201 search space, RandWiredNN search space, and ResNet/ResNeXT search space. The paper is easy to understand and follow. It tackles an important problem that can increase efficiency for a variety of NAS methods.

One of the most interesting general indications from the experimental results thoroughly show that early validation loss is a significantly worse measure of later performance compared to early training loss. Their measure seems to only be beat by validation accuracy quite late in training

Weaknesses:

This paper would benefit from better positioning of the work in relation to related work on performance prediction. While work is well positioned with respect to LcSVR and early validation accuracy in experiments and related work, it would still benefit from additional comparisons to recent work such as tree-based and graph functions like LGBoost[1], XGBoost[2], GIN[3] which were examined as performance predictors in Nasbench-301[4]. LGBoost[1] and XGBoost[2] were also analyzed more thoroughly in [5].

One area it would actually be very beneficial to test would be comparing your TSE metrics with simply using the training loss as a strong baseline for your proposed variations.

An interesting additional area to explore may be how the training loss compares to the training accuracy as a metric.

One area which the paper could be extended to show further robustness of this finding would be testing out a task other than classification such as object detection or regression to see if maybe this is a property limited to simple classification cross entropy loss. If any results could be shown on full sized Imagenet, it would demonstrate that the methodology works also with larger datasets.

It would also be useful to show the robustness of this general method for performance estimation if you could analyze if this can be extended to other hyperparameter search problems as you mentioned.

[1] G. Ke, Q. Meng, T. Finley, T. Wang, W. Chen, W. Ma, Q. Ye, and T. Liu. Lightgbm: A highly efficient gradient boosting decision tree. In I. Guyon, U. V. Luxburg, S. Bengio, H. Wallach, R. Fergus, S. Vishwanathan, and R. Garnett (eds.), Advances in Neural Information Processing Systems 30, pp. 3146–3154. Curran Associates, Inc., 2017.
[2] Julien Siems, Lucas Zimmer, Arber Zela, Jovita Lukasik, Margret Keuper, and Frank Hutter. Nas-bench-301 and the case for surrogate benchmarks for neural architecture search. arXiv preprint arXiv:2008.09777, 2020.
[3] K. Xu, W. Hu, J. Leskovec, and S. Jegelka. How powerful are graph neural networks? In International Conference on Learning Representations, 2019a.
[4] Siems, Julien, et al. "NAS-Bench-301 and the case for surrogate benchmarks for neural architecture search." arXiv preprint arXiv:2008.09777 (2020).
[5] White, Colin, et al. "How Powerful are Performance Predictors in Neural Architecture Search?." arXiv preprint arXiv:2104.01177 (2021).


**Time Spent Reviewing:**

8

---

> ### Author Response · Authors · 2021-08-10
> **Thank you for your comments and our responses are below**
>
> 1.	> *Comparison to LGBoost and XGBoost*
>
> Following your suggestion, we compared our TSE methods to LGBoost and XGBoost on NAS-Bench-201 CIFAR10 and NAS-Bench-301. We tried different numbers of training data (N) for LGBoost and XGBoost (e.g. N=200 means 200 training data) following (Colin et al., 2021) and optimised their hyperparameters for each N by running random search in the ranges proposed in (Siems et. al, 2020) for 1000 iterations. The Spearman’s rank correlation results in the ascending order are shown below. Note TSE-E(T=10) means TSE-E score computed at training epoch 10. It’s evident that our TSE estimators with only T=5 epochs can already outperform LGBoost and XGBoost with N=200 and TSEs with T=10 epochs can outperform LGBoost and XGBoost with N=500.
>
> |**NAS-Bench-201 CIFAR10**||
> | ---  | --- |
> |TSE-E(T=5) | 0.68 |
> |LGBoost(N=200)| 0.71|
> |TSE-EMA(T=5)| 0.72|
> |XGBoost (N=200)| 0.73|
> |LGBoost (N=500)| 0.81|
> |XGBoost (N=500)| 0.83|
> |TSE-E(T=10)| 0.83|
> |TSE-EMA(T=10)| 0.85|
> |**NAS-Bench-301** ||
> |XGBoost (N=200)| 0.43|
> |LGBoost(N=200)| 0.57|
> |XGBoost (N=500)| 0.58|
> |TSE-E(T=5)| 0.68|
> |LGBoost (N=500)| 0.70|
> |TSE-EMA(T=5)| 0.71|
> |TSE-E(T=10)| 0.76|
> |TSE-EMA(T=10)| 0.77|
> ||
>
> Due to rebuttal time constraint, we didn’t implement and run GIN but Siems et. al (2020) has shown on NAS-Bench-301 that both LGBoost and XGBoost can outperform GIN.
>
> 2.	>*Compare with training loss as a baseline*
>
> Thanks for the suggestion. We actually already did that ablation in Fig. 2 (b) to (d)  and Table 2 of the Appendix, where we denote the training loss baseline as TLmini (grey curve). In both cases, TSE clearly outperforms the training loss baseline.
>
> 3.	>*Possible extension: test on tasks other than classification and on full sized ImageNet to show further robustness*
>
> Thanks for the valuable suggestion on potential extensions. A statistically significant evaluation of our proposed estimators on non-classification tasks or on the full sized ImageNet would require fully training and evaluating hundreds of architectures. Thus, it’s not possible to conduct those analyses during the rebuttal period but we agree that they are interesting future directions to look into.
>
> 4.	>*Extended to hyperparameter search problems*
>
> We also agree that extending our proposed performance estimators to hyperparameter optimisation is an interesting future direction.
>
> 5.	>*How the training loss compares to the training accuracy*
>
> We compared them on NAS-Bench-201 in Fig. 3 of the Appendix. Summing training accuracy (SoTAcc) performs very closely to TSE on CIFAR datasets and does slightly worse on ImageNet16-120.

---

> > ### Comment · Reviewer_2hFh · 2021-09-02
> > **Post-rebuttal Update**
> >
> > I would like to thank the authors for answering my questions and running the addition comparison with LGBoost and XGBoost. I believe it is a strong pape.

---

### Official Review · Reviewer_Ctey · 2021-07-23

**Rating:** 6
**Confidence:** 5

**Summary:**

Performance estimation is an important point to speed up the search process in NAS.
This paper proposes a simple method for estimating the generalisation performance of neural architectures via training speed measured by the training loss, which is theoretically motivated by the relation between training speed and generalisation of neural networks.
Experiments demonstrate the effectiveness of the proposed TSE (training speed estimators) compared to other commonly used estimators. Further, it can be incorporated into existing NAS algorithms and make improvements.

**Ethics Review Area:**

["I don’t know"]

**Main Review:**

The proposed TSE is a simple yet effective estimator for performance prediction.
The metric is easy to compute during the training and correlates with the generalisation of neural architectures.
Below are some problems/questions:
1. In line 201, E=1 performs well and implies that more recent training steps is better for TSE. Then what if you limit the minibatches used to calculate the TSE, rather than the B batches in an epoch?
2. In line 268, you choose different N_s values to identify the best T_o. Does this mean that for other practitioners applying your TSE on their task, they can use N_s = 10 to identify the overfit point T_o?
3. In Figure 4, why T=200 is worse than T=10 across all the experiments? This is not reasonable.
4. In line 299, what is the additional cost of training B additional mini-batches?
5. In Figure 1, why does SoVL not appear in d-h, LcSVR not in h, SNIP not in d-h, and JacCov not in d-h ?

Typos and grammar mistakes:
1. Line 28, trainin -> training
2. In Table 1, N_{total} appears in the caption but not in the table, and N_{T} appears in the table rather in the caption.

**Time Spent Reviewing:**

8

---

> ### Author Response · Authors · 2021-08-10
> **Thank you for your helpful comments and our responses are below**
>
> 1.	>*What if you limit the mini-batches used to calculate the TSE below one epoch?*
>
> Following the reviewers’ suggestion, we compute the TSE-E statistics with E < 1 (i.e. within an epoch) on the 20-cell DARTS architecture data (used in Fig. 2 (b) to (d)), for which we have saved each mini-batch training losses. For example, E=0.1 corresponds to the sum of training losses over the last 10% of the mini-batches/optimisation steps in an epoch. On 20-cell DARTS architectures trained with batch size = 96, initial learning rate = 0.025 and cosine learning rate scheduling (setting in Fig. 2 (b)) ,  the Spearman’s rank correlation achieved by TSE-E at T=0.5 * T_end for different E values are:
> - 0.892 (E=0.1) < 0.906 (E=0.3) <  0.910 (E=0.5)  <  0.912 (E=0.7) <  **0.913 (E=1)**.
>
> We see that summing over the entire epoch (E=1) gives the best performance; this might be because E=1 covers the entire training set as Bayesian marginal likelihood and PAC-Bayes bound do, which are the theoretical inspirations for our method. This observation is also consistently observed under the settings in Fig. 2 (c) and (d), and we will update the paper to include curve plots on these results.
>
> 2.	>*When applying your TSE, can we use N_s = 10 to identify the overfit point T_o?*
>
> We would recommend the practitioners to *start with* N_s = 10 when applying our TSE because based on our empirical results on all 3 image tasks on NAS-Bench-20, such a small number of N_s (i.e. N_s = 10) seems to be consistently sufficient to identify the threshold over 100 random trials. In addition, as we highlighted in the paper, in most NAS applications, the training budget T is far below the threshold T_o as T <= 0.5 T_end so it doesn’t require the estimation of T_o in those cases.
>
> 3.	>*Missing baselines in Fig. 1*
>
> We forgot to save the validation loss for Fig. 1(d) so SoVL is missing there. Due to rebuttal time constraint, we only managed to evaluate the top-performing zero-cost proxy, SynFlow, on the task in Fig. 1(d) and it achieves a rank correlation of 0.76, which is lower than 0.89 achieved by TSE-EMA, TSE-E, TSE and VAccES. We will run other zero-cost estimators and update Fig. 1(d) accordingly.
>
> Thanks for spotting out the missing LcSVR in Fig. 1(h). We had added LcSVR to it and will update the paper; LcSVR achieves a best rank correlation of 0.31 at T=0.5 T_end, lower than those achieved by TSE-EMA (0.40), TSE-E (0.40) and VAccES (0.34).
>
> For dataset on (e) to (h), the open-sourced dataset doesn’t contain the exact codes about the search space so we cannot reconstruct the exact ResNe(X)t models to compute zero-cost proxies like SNIP and JacCov.
>
> 4.	>*What is the additional cost of training B additional mini-batches?*
>
> On DARTS search space, if we follow the RandNAS protocol (batch size=64, train/valid split=0.5), **validation cost = 6.6 s** (to compute validation accuracy on entire validation set) and **train cost on B=100 additional mini-batches=7.5 s** for one architecture. Since our TSE doesn’t require the computation of validation statistics, the additional costs incurred compared to the conventional use of validation accuracy for model selection would be 7.5-6.6=0.9s for B=100.
> On NAS-Bench-201 CIFAR10 task, if we follow its protocol, **validation cost=6.4 s** and **train cost on B=100 additional mini-batches=4.4 s**. So we actually reduce time cost when B=100.
>
> 5.	>*Why T=200 is worse than T=10 in Fig. 4?*
>
> In Fig. 4, the x-axis is measuring the run time. So given the same run time, T=200 (full evaluation), which takes ~20 times longer to train each architecture as compared to T=10 (early stopping), will run much fewer search iterations (i.e. search fewer architectures) and thus lead to worse performance. If the x axis is number of architecture queries, T=200 would outperform T=10.
>
> 6.	>*typos*
>
> Thanks for spotting out those typos and we will fix them in the updated version of the paper.

---

> > ### Comment · Reviewer_Ctey · 2021-08-18
> > **Thanks for your reply**
> >
> > Thanks for your reply on the questions.

---

> > > ### Author Response · Authors · 2021-08-25
> > > **Thanks for your comments**
> > >
> > > We are glad that our responses address your questions. If we have addressed them well, we would very much appreciate it if you could consider increasing your score. Thanks for your time and efforts reviewing our work and for your valuable comments which will help improve our paper.

---

### Decision · Program_Chairs · 2021-09-27

**Decision:**

Accept (Spotlight)

**Comment:**

This is a strong paper in a somewhat crowded research area, performance prediction for NAS. Even within such a crowded area, this paper stands out for the thoughtfulness of its experiments and the grounding of the approach proposed. The reviewers asked for more depth in the related work section and for some additional experiments, but overall have no remaining concerns regarding the paper.